# Post-translational covalent assembly of CAR and synNotch receptors for programmable antigen targeting

Elisa Ruffo [1,2,3,4], Adam A. Butchy [5], Yaniv Tivon[6], Victor So [1,2,3,4], Michael Kvorjak [1,2,3,4], Avani Parikh [1,2,3,4], Eric L. Adams[1,2,3,4], Natasa Miskov-Zivanov [5,7,8], Olivera J. Finn[3], Alexander Deiters [6] & Jason Lohmueller [1,2,3,4] ✉

Chimeric antigen receptors (CARs) and synthetic Notch (synNotch) receptors are engineered cell-surface receptors that sense a target antigen and respond by activating T cell receptor signaling or a customized gene program, respectively. Here, to expand the targeting capabilities of these receptors, we develop "universal" receptor systems for which receptor specificity can be directed post-translationally via covalent attachment of a co-administered antibody bearing a benzylguanine (BG) motif. A SNAPtag self-labeling enzyme is genetically fused to the receptor and reacts with BG-conjugated antibodies for covalent assembly, programming antigen recognition. We demonstrate that activation of SNAP-CAR and SNAP-synNotch receptors can be successfully targeted by clinically relevant BG-conjugated antibodies, including anti-tumor activity of SNAP-CAR T cells in vivo in a human tumor xenograft mouse model. Finally, we develop a mathematical model to better define the parameters affecting universal receptor signaling. SNAP receptors provide a powerful strategy to post-translationally reprogram the targeting specificity of engineered cells.

Engineered antigen receptors are revolutionizing the treatment of blood cancers and show promise in cell therapies treating a wide range of other diseases[1]. The most clinically advanced of these technologies are chimeric antigen receptors (CARs), synthetic T cell receptors most often comprised of an antigen-specific antibody single chain variable fragment (scFv) fused by spacer and transmembrane domains to intracellular T cell signaling domains[2–4]. Upon binding to a target antigen, CARs stimulate T cell activation and effector functions including cytokine production, cell proliferation, and target cell lysis. Adoptively transferred CAR T cells targeting the B cell antigen CD19 and BCMA are now FDA-approved and have been highly successful in

treating refractory acute lymphoblastic leukemia, large B cell lymphoma, mantle cell lymphoma, follicular lymphoma, and multiple myeloma[5–8]. Creating CARs against additional targets to treat other types of cancer and immune-related diseases is a major research focus[9,10]. Another class of highly versatile antigen receptors are synthetic Notch (synNotch) receptors which consist of an antigen binding domain, the Notch core protein from the Notch/Delta signaling pathway, and a transcription factor[11–13]. Instead of activating T cell signaling upon binding to the target antigen, the Notch core protein is cleaved by endogenous cell proteases thus releasing the transcription factor from the cell membrane. Subsequent nuclear translocation leads to

[1]UPMC Hillman Cancer Center, University of Pittsburgh, Pittsburgh, PA, USA. [2]Division of Surgical Oncology, Department of Surgery, University of Pittsburgh, Pittsburgh, PA, USA. [3]Department of Immunology, University of Pittsburgh, Pittsburgh, PA, USA. [4]Center for Systems Immunology, University of Pittsburgh, Pittsburgh, PA, USA. [5]Department of Bioengineering, University of Pittsburgh, Pittsburgh, PA, USA. [6]Department of Chemistry, University of Pittsburgh, Pittsburgh, PA, USA. [7]Department of Electrical and Computer Engineering, University of Pittsburgh, Pittsburgh, PA, USA. [8]Department of Computational and Systems Biology, University of Pittsburgh, Pittsburgh, PA, USA. ✉e-mail: lohmuellerj@upmc.edu

transcriptional regulation of one or more target genes. These receptors are highly modular as they can be created to target different cell surface antigens by changing the scFv, and they can positively or negatively regulate any gene of interest by either fusing different transcription factors as components of the receptors or by changing the transgenes under their control. This versatile receptor type is of great clinical interest in immunotherapies as well as applications to tissue engineering[14–16].

To gain additional control over CAR function, we and others have developed "universal" adaptor CAR systems for which the CAR, instead of directly binding to an antigen on a target cell, binds to a common tag molecule fused or conjugated to an antigen-specific antibody[17–23]. These systems are designed such that a patient is infused with both a tagged, antigen-specific antibody adaptor that binds to target cells and CAR T cells that become activated by the tagged antibody at the surface of target cells. Adaptor CARs are referred to as "universal CARs" as they have the potential to allow for one population of T cells to target multiple tumor antigens by administering different antibodies sequentially or simultaneously. Additionally, the activity of the adaptor CARs can be tuned by altering the concentration of tagged antibodies, administering the tag molecule as a competitive inhibitor, or halting antibody administration for better control over potential toxicities resulting from over-active CAR T cells. Adaptor CAR systems that recognize a variety of peptides or small molecules conjugated to antibodies have been developed, including biotin, fluorescein, peptide neo-epitopes (PNE), Fcγ, and leucine zippers[17–24]. Several adaptor CAR systems are currently being tested in clinical trials.

Here, we describe key advances in antigen receptor design – the creation of a universal adaptor synNotch system and the creation of a universal CAR system that both act through self-labeling enzyme chemistry. Our first attempt to create an adaptor synNotch system that functioned through transient binding of the receptor to an antibody was unsuccessful, and we reasoned that a stronger antibody-receptor interaction would be necessary. Seeking to create a synNotch receptor with a stronger interaction, we generated a synNotch receptor containing the SNAPtag protein designed to covalently fuse to the adaptor antibody. SNAPtag is a modified human O-6-methylguanine-DNA methyltransferase (MGMT) that was engineered to react to benzylguanine, a bio-orthogonal tag molecule, and is known to be specific and efficient at self-labeling (meaning that it will perform a chemical reaction to covalently fuse with molecules containing the BG motif) in both cells and animals (Fig. 1a)[25–27]. As previous reports for adaptor CARs have shown a positive correlation between CAR function and the CAR-adaptor binding affinity, we also created and characterized a CAR containing the SNAPtag[17,20]. The SNAP-CAR and SNAP-synNotch systems are highly modular receptor platforms for diverse programming of cell behaviors using covalent chemistry (Fig. 1b–d).

## Results
### Engineering a self-labeling SNAP universal adaptor synNotch receptor
We created a synNotch receptor containing the SNAPtag self-labeling enzyme which forms a covalent bond via cysteine benzylation with a BG-tagged molecule and the Notch core protein fused with a

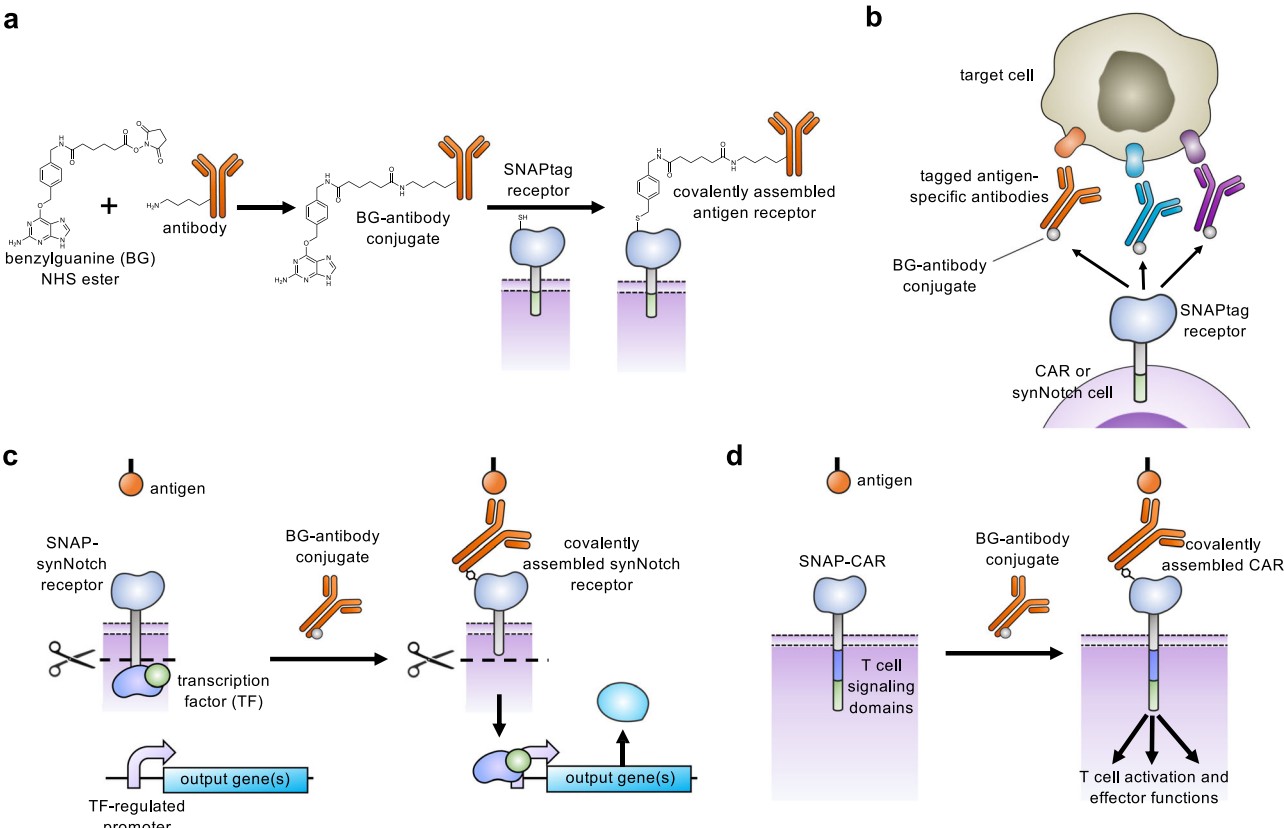

**Fig. 1 | Universal adaptor SNAP-CAR and SNAP-synNotch receptor function. a** A benzylguanine motif (BG) is chemically conjugated to an antibody using a benzylguanine NHS ester. The BG-antibody conjugate then covalently binds to the extracellular SNAPtag enzyme through a self-labeling reaction. **b** SNAPtag receptors enable the targeting of multiple different antigens using the same receptor by combining SNAP receptor cells with different BG-conjugated antibodies. **c** The SNAP-synNotch receptor is targeted by a BG-conjugated antibody and upon antigen recognition leads to cleavage of the synNotch receptor, releasing the transcription factor and transcriptional regulation of a target gene or genes. **d** The SNAP-CAR is targeted by a BG-conjugated antibody to activate T cell signaling and effector functions upon antigen recognition.

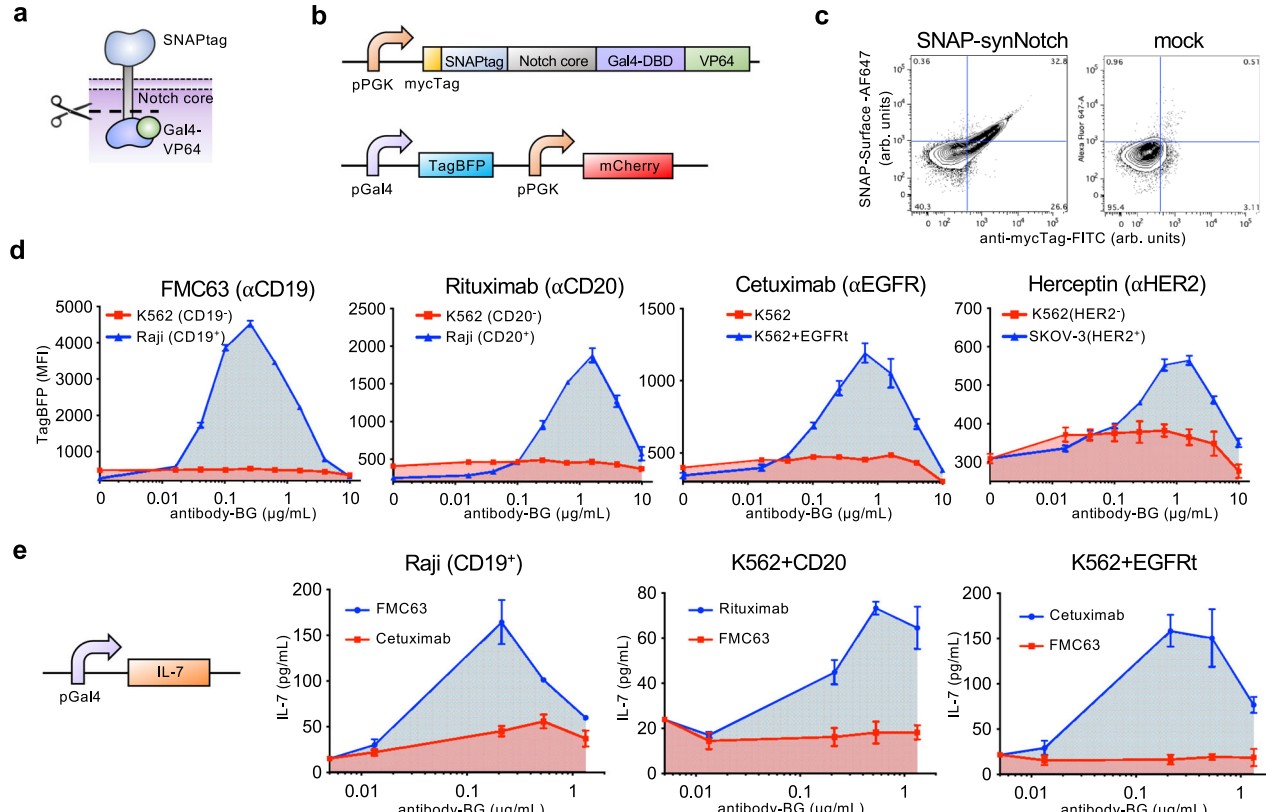

**Fig. 2 | The SNAP-synNotch receptor can be targeted to desired antigens of interest by benzylguanine-conjugated antibodies. a** Diagram of the SNAP-synNotch receptor. **b** Lentiviral vector design for SNAP-synNotch receptor expression and the corresponding reporter gene. The SNAP- synNotch receptor contains the Gal4-VP64 transcription factor which upon activation leads to activation of the TagBFP reporter. **c** Flow cytometry analysis of surface expression and enzymatic functionality of the SNAP-synNotch receptor on transduced vs. mock (non-transduced) Jurkat cells assessed by staining with an anti-mycTag antibody and a SNAP-Surface-AF647 dye (arbitrary units, arb. units). **d** Flow cytometry analysis of the activation of SNAP-synNotch Jurkat cells co-incubated with the indicated target cell lines and antibody concentrations for TagBFP output gene expression reported as mean fluorescence intensity (MFI) gated on mCherry+ cells and **e** by ELISA for the production of the IL-7 therapeutic transgene. For **d** and **e**, $n = 3$ biologically-independent experiments ± s.e.m. Source data are available as a Source Data file.

Gal4-VP64 transcription factor (Fig. 2a)[26–28]. The goal of this system was to direct antigen-specific receptor activation by combining SNAP-synNotch cells with BG-labeled antibodies (Fig. 1b, c). In brief, the process of synNotch receptor activation involves antigen binding, which leads to mechanical pulling forces that stretch the receptor and expose proteolytic cleavage sites in the Notch core protein, culminating in the release of the Gal4-VP64 transcription factor from the membrane and transcriptional activation of the Gal4 target gene, the TagBFP reporter in our system. We generated a lentiviral vector encoding the SNAP-synNotch receptor and transduced Jurkat cells (Fig. 2b). Antibody labeling of the myc epitope tag, labeling with a fluorophore-conjugated BG reagent, and analysis by flow cytometry confirmed cell surface expression of the receptor and SNAP-BG cell-surface labeling activity (Fig. 2c).

To generate the adaptor antibodies, we conjugated BG to lysines or N-termini of several clinically relevant antibodies using a synthetic BG-NHS ester (Fig. 1a). These antibodies included Rituximab targeting CD20, FMC63 targeting CD19, Herceptin targeting HER2, and Cetuximab targeting EGFR[29–32]. While the conjugation products were heterogenous, we quantified the average number of BG molecules conjugated to each antibody by a SNAPtag protein-labeling assay in which SNAPtag-conjugation led to a shift in the antibodies' molecular weight that could be resolved by SDS-PAGE (Supplementary Fig. S1). The frequency of BG molecules per antibody ranged from 2.0–2.8 as summarized in Supplementary Table S1. We characterized the antigen expression and BG-antibody staining of various target cell lines by flow

cytometry in which the antibodies displayed expected antigen specificities (Supplementary Fig. S2).

Next, we tested the BG-conjugated FMC63 antibody (FMC63-BG) for its ability to activate synNotch signaling in response to CD19-positive tumor cells (Fig. 2d & Supplementary Fig. S3). We performed a co-incubation assay of SNAP-synNotch Jurkat cells and CD19 positive and negative tumor cells in the presence of different levels of the FMC63-BG antibody conjugate, and after 48 h we assayed for TagBFP reporter gene expression by flow cytometry. TagBFP expression was significantly upregulated in response to CD19-positive tumor cells for various concentrations of antibody. Receptor activation was sensitive, with significant activation observed at an adaptor concentration as low as 0.04 µg/mL and increasing to a peak at 0.25 µg/mL. Reporter gene activation then decreased with increasing antibody amounts before being completely inhibited at a concentration of 10 µg/mL, indicative of a "hook effect". In this case, the antibody is in such excess that different antibody molecules are saturating both the target cells and synNotch cells without the formation of ternary complexes. This behavior is commonly observed with chemical and cell processes that involve ternary complex formation such as receptor/ligand interactions[33]. It is accounted for mechanistically by saturating amounts of ligand binding two surface receptors blocking the formation of ternary complexes[34, 35]. Overall, these experiments confirm the tunability of synNotch activation through careful adaptor titration, even allowing for the dampening of activity at high concentrations.

We found that BG-conjugated antibodies targeting other antigens were also capable of activating the SNAP-synNotch receptor in an

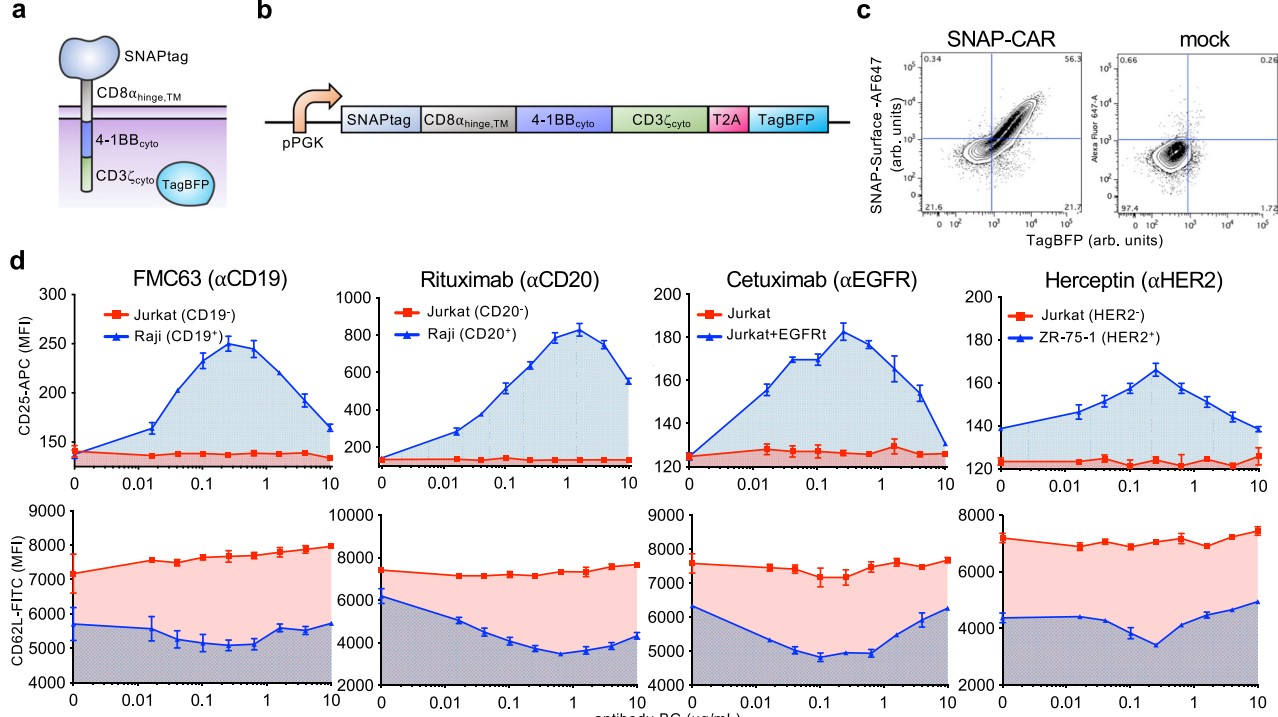

**Fig. 3 | The SNAP-CAR can be targeted to desired antigens of interest by benzylguanine-conjugated binding proteins. a** SNAP-CAR design. **b** SNAP-CAR lentiviral expression construct. **c** Flow cytometry analysis of the expression and enzymatic functionality of the SNAP-CAR receptor on transduced vs. mock (non-transduced) Jurkat cells, assessed by staining with SNAP-Surface-AF647 dye and recording TagBFP expression (arbitrary units, arb. units). **d** Flow cytometry analysis of CD25 and CD62L T cell activation markers on Jurkat SNAP-CAR effector cells (gated by TagBFP+ expression) co-incubated with the indicated target cell lines and antibody concentrations reported as mean fluorescence intensity (MFI). CD25 increases while CD62L decreases with activation, $n = 3$ biologically independent experiments; averages ± s.e.m. Source data are available as a Source Data file.

antigen-specific manner (Fig. 2d). We performed similar co-incubation assays of SNAP-synNotch Jurkat cells and antigen-positive and negative tumor cells in the presence of different levels of BG-conjugated Cetuximab, Herceptin, and Rituximab antibodies and assayed for TagBFP gene expression by flow cytometry. Significant up-regulation of TagBFP was again observed for each of the tested antibodies, tunable through increasing antibody concentrations, and in a target antigen-specific manner. A "hook effect" was observed for each of the antibodies, showing an increase in activity with increasing antibody dose until a peak activation level is reached, followed by an antibody dose-dependent decrease. Receptor activation was also dependent on the number of target cells, having optimal activity at high target to SNAP-synNotch cell ratios (Supplementary Fig. S4).

Next, we tested whether the SNAP-synNotch receptor was capable of regulating the expression of the IL-7 response gene, a candidate therapeutic gene of interest for its ability to promote T cell proliferation[36]. We generated an IL-7 response gene expression construct in which the TagBFP gene was replaced by the IL-7 coding region and placed it under synNotch control[36]. This construct was to again be transcriptionally activated by the Gal4-VP64 transcription factor upon receptor activation. We transduced SNAP-synNotch Jurkat cells with this response vector and co-incubated them with several different antibodies and antigen positive and negative tumor cells for evaluation of IL-7 response gene expression by ELISA (Fig. 2e). Similar to TagBFP response gene activation, IL-7 was significantly up-regulated in an antigen-specific and antibody dose-responsive manner, demonstrating the modularity of the SNAP-synNotch system in controlling different output genes.

Of note, we had first created and tested a putative universal synNotch system with a biotin-avidin tag-receptor interaction using the monomeric biotin-binding protein mSA2 as the

targeting domain[19]. The goal of this system was to target receptor specificity by combining mSA2-synNotch cells with biotinylated antibodies, analogous to our previously reported mSA2 CAR T cell system (Supplementary Fig. 5a)[19]. While we found that the receptor was efficiently expressed on the cell surface and could be activated by plate-immobilized biotin (Supplementary Fig. 5b, c) the receptor was ultimately not functional at detecting cell-surface antigen, as we saw no activation when we incubated the cells with biotinylated antibody-labeled tumor cells (Supplementary Fig. 5d). We posited that the lack of signaling by the mSA2-synNotch receptor, in contrast to potent signaling by mSA2 CAR T cells, was the result of the Notch receptor's differing signaling mechanism that requires a pulling force. We reasoned that a stronger receptor-to-tag interaction (mSA2-biotin $K_d = 5.5 \times 10^{-9}$ M) such as the covalent and thus permanent bond of SNAPtag to BG would be required to create a functional, universally applicable synNotch system[37,38].

### Engineering a self-labeling SNAP universal adaptor CAR
We next aimed to create a universal adaptor CAR system using the SNAPtag protein domain to target T cell receptor signaling when combined with BG-tagged antibodies (Fig. 1d). We cloned the SNAPtag domain into a lentiviral vector containing the CD8α hinge and transmembrane domains, the 4-1BB cytoplasmic domain, the CD3zeta T cell cytoplasmic domain, and a TagBFP reporter gene co-expressed via the T2A co-translational peptide (Fig. 3a, b). We packaged this vector into lentiviral particles and transduced Jurkat cells. We found that the receptor was efficiently expressed and that the SNAPtag protein was functional, as indicated by TagBFP expression and staining with a BG-AF647 fluorophore reagent by flow cytometry (Fig. 3c).

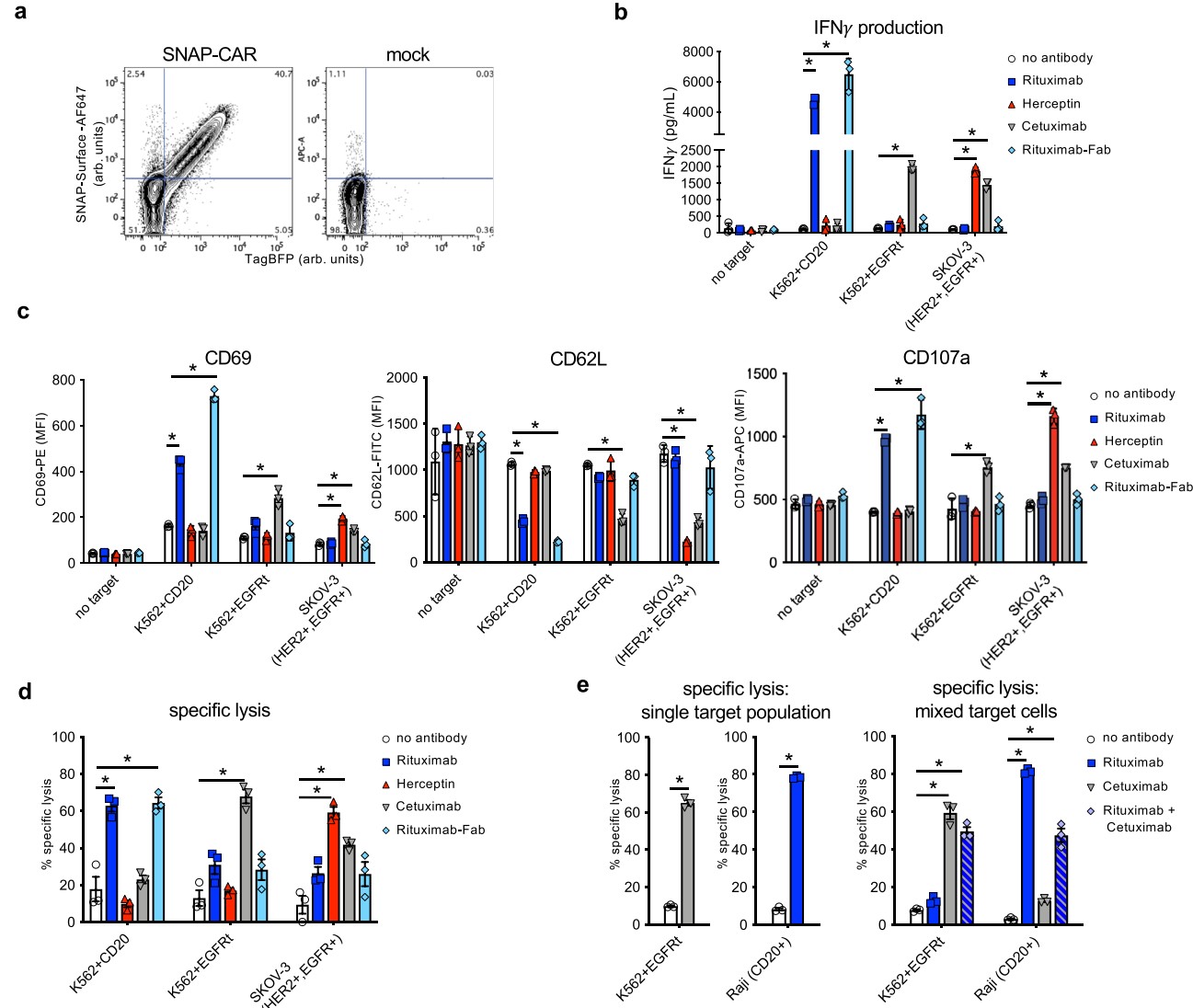

**Fig. 4 | The SNAP-CAR is effective on primary human T cells. a** Flow cytometry analysis of the expression and enzymatic functionality of the SNAP-CAR on transduced vs. mock (un)transduced primary human T cells by staining with SNAP-Surface-AF647 dye and recording TagBFP expression (arbitrary units, arb. units). **b** ELISA for IFNγ production from primary human SNAP-CAR T effector cells co-incubated with the indicated target cell lines and 1.0 μg/mL of the indicated antibody and **c** flow cytometry analysis of CD69, CD62L, and CD107a T cell activation markers on the SNAP-CAR (TagBFP + ) population from the co-incubations in **b** reported as MFI. **d** Specific lysis of target cell lines by co-incubated primary

human SNAP-CAR T cells and 1.0 μg/mL of the indicated BG-conjugated antibodies. **e** Specific lysis of individual cell lines (left) and mixed cell lines (right) by primary human SNAP-CAR T cells and 1.0 μg/mL of the indicated BG-conjugated antibodies. For **b**–**e** two-way ANOVA tests with multiple comparisons were performed. As the data did not have homogeneity of variance (Levene's test), Tukey's HSD was used for post-hoc analysis between antibody conditions. " * "denotes a significance of $p < .0001$, $n = 3$ biologically-independent experiments ± s.e.m. Source data are available as a Source Data file.

Next, we tested whether BG-conjugated antibodies could be combined with SNAP-CAR Jurkat cells to target their T cell activation signaling. We co-incubated SNAP-CAR cells with various antigen positive or negative tumor cell lines and increasing doses of BG-conjugated antibodies. After 24 h we assayed for T cell activation by staining with antibodies specific for CD25, which is up-regulated upon T cell activation, and CD62L, which is down-regulated. We assessed the expression levels of these markers in SNAP-CAR cells by specifically gating on the TagBFP CAR + cell population. We found that expression of these markers was controlled in an antigen-specific and dose-responsive manner by the BG-conjugated antibodies (Fig. 3d). Similar to SNAP-synNotch cells, SNAP-CAR activation signaling strength peaked at an antibody dose level between 0.1–1.0 μg/mL before steadily decreasing, again indicative of a hook effect.

## SNAP-CAR is functional in primary human T cells

We then assessed the expression level and in vitro functional activity of the SNAP-CAR in primary human T cells transduced with the SNAP-CAR lentivirus. Staining with a BG-AF647 fluorophore conjugate and assaying by flow cytometry, we found that the SNAP receptor was efficiently expressed in ~40% of cells in a manner that correlated well with the expression of the TagBFP marker gene (Fig. 4a).

To test CAR functionality, we co-incubated SNAP-CAR T cells with various antigen positive or negative target tumor cell lines and 1.0 μg/ mL of BG-conjugated antibodies for 24 h. Targeted antigens included CD20, EGFR, and HER2. Analyzing the supernatants of the co-incubated cells by ELISA we found that the SNAP-CAR T cells could be directed by the covalently attached BG-antibodies to produce significant amounts of IFNγ in response to antigen positive target cells (Fig. 4b). Our analysis of co-incubated cells by flow cytometry revealed

that the antibodies also led to induction of T cell activation markers, up-regulation of CD69 and CD107a, and down-regulation of CD62L (Fig. 4c). CAR T cells and target cells were also co-cultured and evaluated for target cell lysis by flow cytometry, and high levels of target cell specific lysis were observed (Fig. 4d). Lysed target cells were identified by staining with the Ghost Dye fluorescent dye gating on the CellTrace positive (target cell) population. Again, T cell marker activation and target cell lysis were significantly higher only when the co-administered antibody targeted an antigen expressed by the co-administered cells, indicating antibody specificity. In addition to full-length IgG antibodies, we also tested a BG-conjugated Fab fragment of Rituximab. This molecule, more similar to the scFv antibody fragment found in traditional CARs, also showed potent activity for each effector function equal to or greater than the full-length Rituximab.

To explore a potential method to further tune SNAP-CAR activity, we tested whether the adaptor tag molecule alone, O6-benzylguanine (O6-BG), could act as a competitive inhibitor. O6-BG was developed as an anti-neoplastic agent, acting as a suicide inhibitor for the O6-methylguanine-DNA methyltransferase DNA repair protein[39]. Importantly, when tested in clinical trials targeting glioma, it was found to lead to no toxicities even at the highest doses tested. Co-incubating SNAP-CAR T cells, target cells, and antibody adaptors, with titration of O6-BG showed that O6-BG could compete with antibody adaptor and inhibit SNAP-CAR activity in a dose-dependent manner. The killing ability of the SNAP-CAR system was completely abolished at $0.1\,\mu M$ of O6-BG. Moreover, T cell activation markers such as CD69 and CD107a, were reduced along with CD62L marker increase, demonstrating the ability of O6-BG to competitively inhibit SNAP-CAR function (Supplementary Fig. 6).

Next, we sought to test whether SNAP-CAR T cells could simultaneously target multiple antigens on different cell populations demonstrating the potential to prevent cancer relapse due to antigen heterogeneity or antigen loss. We co-cultured SNAP-CAR T cells with a mixture of CD20+ and EGFR+ target cells and assayed for cell lysis with BG-modified Rituximab, Cetuximab, or the combination of both antibodies. With single antibodies we observed specific lysis of each individually targeted cell line within the mixed cell population. When both antibodies were combined, the SNAP-CAR T cells mediated significant, simultaneous lysis of both targeted cell lines (Fig. 4e).

### BG-antibody pre-loading experiments interrogate the receptor signaling mechanism

To further investigate the mechanism of SNAP-synNotch receptor activation, we performed co-incubation assays in which we pre-labeled either the target cells or the SNAP-synNotch Jurkat cells with BG-conjugated antibodies. We incubated CD19 positive and negative tumor cells with BG-conjugated antibody, washed away residual unbound antibody, and co-incubated these cells with SNAP-synNotch cells for 48 h. Evaluating response gene activation, we found that CD19 positive tumor cells significantly up-regulated TagBFP gene expression to a level comparable to the peak activation level observed in previous dose-response assays (Figs. 5a and 2c). When we instead pre-labeled the SNAP-synNotch cells with BG-conjugated antibody and performed a similar co-incubation assay, no significant response gene activation was observed (Fig. 5b).

Performing the same pre-staining experiments with SNAP-CAR cells, we found that the SNAP-CAR was functional both when the SNAP-CAR cells or tumor cells were pre-labeled with BG-conjugated antibodies. Antigen positive or negative tumor cells were labeled with BG-conjugated antibody, washed, and co-incubated with Jurkat SNAP-CAR cells for 24 h. Assaying by flow cytometry, we saw that the labeled tumor cells induced a significant up-regulation of CD25 expression to a level comparable to the peak level of activation in the previous dose-response assay (Fig. 5a). When pre-labeling the SNAP-CAR cells with BG-labeled antibody, significant up-regulation of T cell activation was

observed (Fig. 5b). Finally, we also tested the killing capacity of adaptor pre-loaded primary human SNAP-CAR T cells. We pre-labeled SNAP-CAR T cells with different concentrations of Rituximab-BG adaptor and at varying cell concentrations. We assayed a subset of these cells for adaptor loading by staining with a fluorescently labeled anti-human IgG antibody, and we observed concentration-dependent labeling of antibody by flow cytometry (Supplementary Fig. 7). We then incubated these cells with CD20 positive target cells for 24 h, and assayed target cells for lysis (Fig. 5c). We found that the pre-labeled SNAP-CAR T cells significantly induced specific lysis in an antibody-dose dependent manner. This level was lower but comparable to that of cells incubated with a soluble adaptor and a conventional anti-CD20 CAR. T cell activation markers also correlated with adaptor dose (Supplementary Fig. 8a, b).

### SNAP-CAR can be labeled by antibody adaptors in vivo

To test the therapeutic feasibility of the SNAP-CAR T cell approach, we evaluated whether SNAP-CAR T cells could be labeled with antibody adaptors in vivo in mice. First, to generate a more clinically appropriate expression system, we made a gammaretroviral expression vector containing the SNAP-CAR gene co-expressed with the LNGFR marker gene via the T2A co-translation peptide (Fig. 6a)[40]. Primary human T cells transduced with this construct displayed high levels of CAR expression as evidenced by LNGFR staining (Fig. 6b). Next, to test in vivo loading of the SNAP-CAR with antibody adaptors, we administered these cells via retro-orbital (r.o.) injection into NOD-SCID-γchain-deficient (NSG) mice and then administered either PBS or antibody adaptor via intraperitoneal (i.p.) injection. After 24 h we collected blood via the submandibular vein and isolated lymphocytes, and stained for SNAP-CAR expression using an anti-LNGFR antibody and adaptor labeling via staining with a fluorescently labeled anti-human IgG antibody. Mice injected with antibody adaptors displayed significantly higher anti-human IgG staining that correlated with the level of LNGFR expression (Fig. 6c, d).

### SNAP-CAR T cells show anti-tumor function in vivo in a human tumor xenograft mouse model

To test the therapeutic potential of our SNAP-CAR approach, we evaluated SNAP-CAR T cells for anti-tumor activity in a human tumor xenograft mouse model. To create a system to compare SNAP-CAR activity to a traditional CAR in vivo, we generated anti-HER2 CAR T cells (scFv from mAb 2C4) and confirmed anti-HER2 function in vitro (Supplementary Figure S9). NSG mice were injected intravenously (i.v.) with NALM6 + HER2 tumor cells that express firefly luciferase. After 4 days, mice underwent IVIS imaging for tumor burden, and then were divided into four treatment groups: adaptor alone, SNAP-CAR T cells only, SNAP-CAR T cells plus anti-HER2 adaptor, and anti-HER2 CAR T cells (Fig. 7a). CAR T cells were injected i.v. on day 4, and the adaptor was administered i.p. every 3 days. Prior to injection, CAR T cells were evaluated for CAR expression, as well as T cell phenotype markers CD4, CD8, and CD62L (Supplementary Fig. 10). Of note, intravenous immunoglobulin (IVIG) was administered i.p. on day 4 with CAR T cells and with every adaptor injection to enhance SNAP-CAR T cell engraftment[41]. In previous testing, lack of IVIG led to a reduction of ~50% of SNAP-CAR T cells in the blood of mice after only 24 h (Fig. 6c and Supplementary Fig. 11). We hypothesize that the IVIG is protecting SNAP-CAR cells from Fc Receptor interactions with innate immune cells or stromal cells that are occurring due to lack of circulating antibodies in the NSG mouse model. Mice underwent IVIS imaging every 5 days until day 40 and then at day 60 for tumor burden. While adaptor-only and SNAP-CAR T cell only controls showed rapid tumor growth, both SNAP-CAR T cells with adaptor and anti-HER2 traditional CAR T cell treatment groups showed significant inhibition of tumor growth in all mice and had 4/5 mice showing no signs of tumor growth at day 60. To further demonstrate the reproducibility of these results,

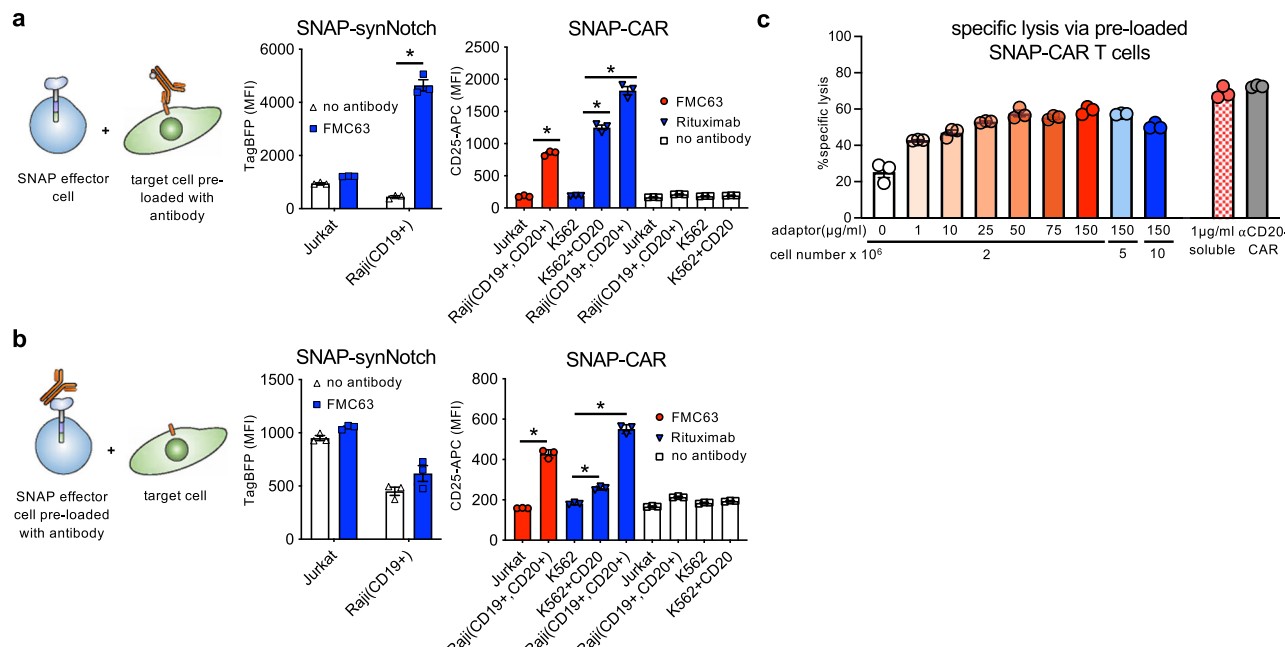

**Fig. 5 | Characterizing the activity of SNAP receptors when pre-assembled or with pre-labeled target cells. a** Flow cytometry analysis of SNAP receptor activation for SNAP-synNotch and SNAP-CAR cells co-incubated with target cells that were pre-labeled with the indicated antibodies. **b** Flow cytometry analysis of SNAP receptor activation for SNAP-synNotch and SNAP-CAR cells that were pre-labeled with the indicated antibodies and co-incubated with target cells. TagBFP output gene expression and CD25 marker expression were evaluated by flow cytometry. **c** Specific lysis of target cells by co-incubated primary human SNAP-CAR T cells that were pre-incubated with the indicated concentration of adaptor at the indicated cell concentration as compared to SNAP-CAR T cells incubated with 1.0 μg/mL of soluble adaptor or a positive control anti-CD20 CAR. For **a** and **b**, two-way ANOVA tests with multiple comparisons were performed. As the data did not have homogeneity of variance (Levene's test), Tukey's HSD was used for post-hoc analysis between antibody conditions. " * " denotes a significance of $p < 0.0001$, $n = 3$ biologically-independent experiments ± s.e.m. For **c** a one-way ANOVA with Dunnet's Multiple Comparison tests was performed and all values were significantly different ($p < 0.0001$) from the no adaptor control (white bar). Source data are available as a Source Data file.

the experiment was repeated using SNAP-CAR T cells generated using cells isolated from a different donor (Supplementary Fig. 12). SNAP-CAR T cells with adaptor again showed significant inhibition of tumor growth compared to the control groups with 5/5 mice showing no signs of tumor growth. Notably, at day 35 we observed SNAP-CAR T cells present in the blood of mice, showing CAR positivity (LNGFR + ) close to that of the injected cell population, indicating persistence of these cells to at least day 35 (Supplementary Fig. S13).

In order to further demonstrate the targeting versatility of the SNAP-CAR system in vivo, we investigated the ability to mediate anti-tumor activity against an additional antigen, CD20, via the Rituximab-BG adaptor. We constructed a NALM6 cell line expressing CD20, NALM6 + CD20, and we generated and confirmed in vitro activity of anti-CD20 CAR T cells (scFv from mAb leu16) (Fig. 5c and Supplementary Fig. S8)[42]. Prior to in vivo experimentation, CAR T cells were evaluated for CAR expression via antibody staining for LNGFR, as well as T cell subset and memory phenotype markers CD4, CD8, CD62L, and CD45RA. This analysis showed high similarity in CAR expression and memory T cell phenotypes between the SNAP-CAR T cells and the traditional anti-CD20 CAR T cells (Supplementary Fig. S14). NSG mice were inoculated with NALM6 + CD20 tumor cells. CAR T cells and Rituximab-BG adaptor with IVIG were administered and mice were imaged using the same dosages and timeframe of the anti-HER2 experiments described above. While adaptor-only and SNAP-CAR T cell-only controls showed rapid tumor growth, both SNAP-CAR T cells with adaptor and traditional anti-CD20 CAR T cell treatment groups showed significant inhibition of tumor growth in all mice (Supplementary Fig. S15). While 5/5 mice treated with SNAP-CAR T cells + Rituximab adaptor as well as 3/5 anti-CD20 CAR mice ultimately showed only partial responses (PR) with tumor relapse, upon investigating CD20 antigen expression of tumor cells in the blood, we

observed that the NALM6 cells (identified in the mouse blood by human CD19 expression) had lost CD20 antigen expression suggesting that the eventual outgrowth of tumor cells was not due to a defect in CD20-targeting (Supplementary Fig. S16). Looking at CD20 expression of the injected NALM6 + CD20 cells, we observed a small starting population of CD20(-) cells, and we hypothesize that these cells were able to avoid CAR targeting and expand over time in the mice (Supplementary Fig. S16). Similar to the experiments targeting HER2, CAR positive cells were present in the mice at a late time point (day 40) (Supplementary Fig. S17).

**Mathematical model of universal adaptor complex formation**
To gain a better understanding of universal adaptor receptor signaling and the observed hook effect, we generated a continuous mathematical model of the ternary complex formation between T cells, adaptor antibodies, and target cells. Using Python Jupyter Notebook, we created a generalizable model of ordinary differential equations (ODEs) that describes the binding reactions. A system of equations was defined to describe the accumulation and concentration of each of the six species in the model: T cells, antibodies, tumor cells, T cells bound to antibody, tumor cells bound to antibody, and T cell-antibody-tumor cell ternary complexes (Fig. 8a, Methods).

To validate the model, we ran simulations using kinetic parameters taken directly from the literature and then through bounded parameter fitting (Supplementary Table S2)[43–46]. Using direct literature values, the model was able to recapitulate the general features of our experimental data, including the observed hook effect and a prediction accurate within an order of magnitude of each antibody dose expected to yield maximum receptor signaling (Fig. 8b). We used the sum of squared error (SSE) calculations to measure the error in model

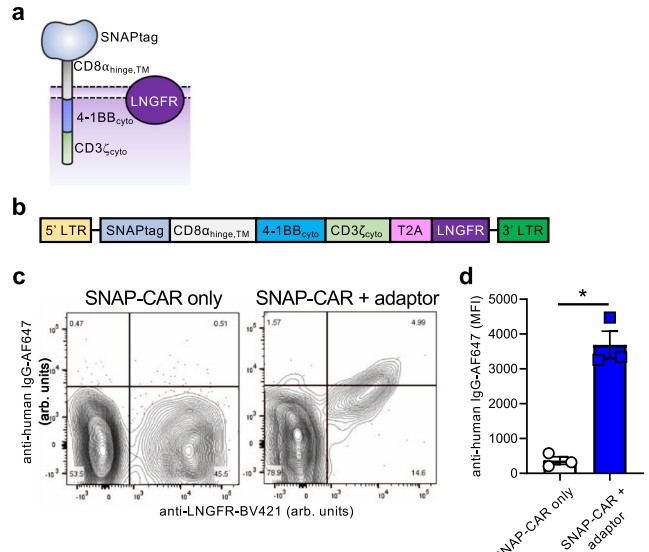

**Fig. 6 | In vivo loading of SNAP-CAR T cells with antibody adaptor. a** Diagram of the SNAP-CAR-LNGFR receptor. **b** Design of the SNAP-CAR gammaretroviral expression construct. **c** Flow cytometry analysis of SNAP-CAR T cells from the blood of NSG mice injected r.o. 24 h prior with SNAP-CAR T cells with or without Rituximab antibody adaptor i.p. (arbitrary units, arb. units). **d** Average MFI of anti-human IgG-AF647 on LNGFR + cells. For **d** an unpaired two-tailed student's t-test was performed and "*" denotes a significance of $p = 0.012$, $n = 3$ mice ± s.e.m. Source data are available as a Source Data file.

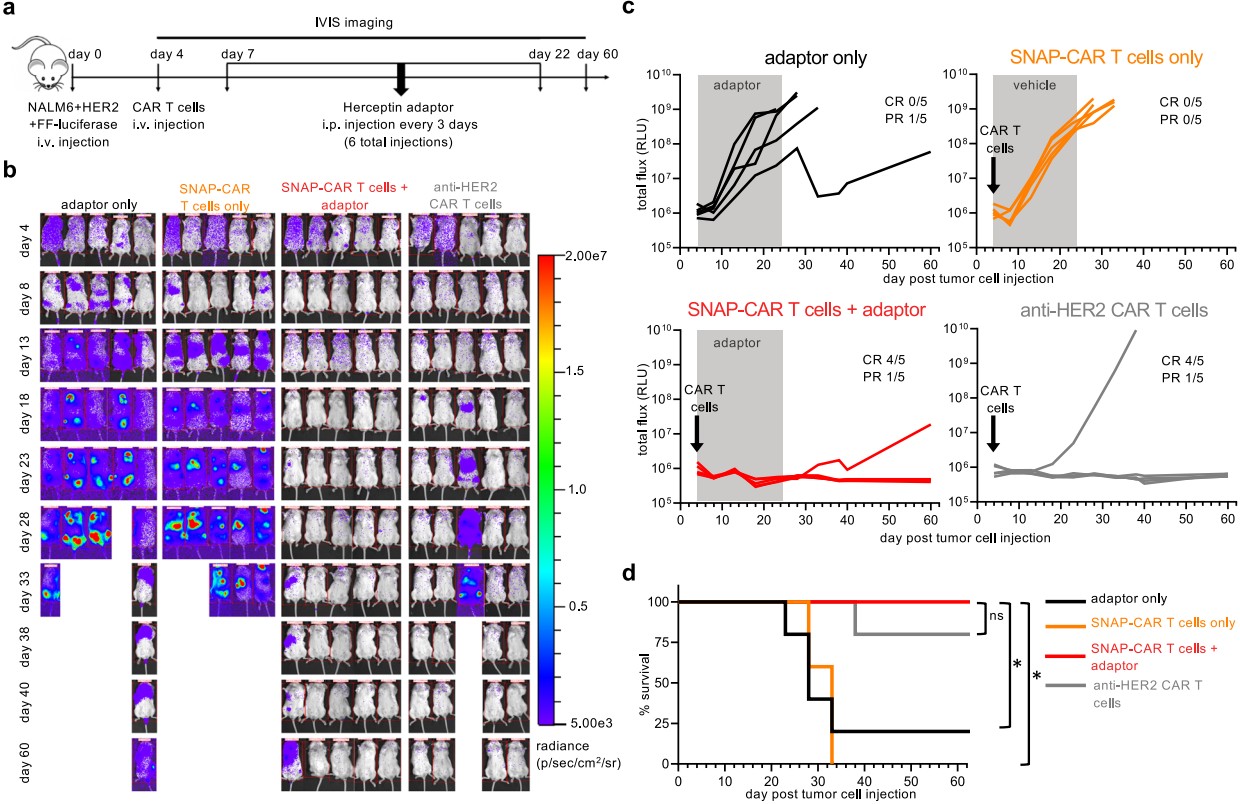

**Fig. 7 | Anti-tumor activity of SNAP-CAR T cells in vivo in a human tumor xenograft mouse model. a** In vivo experimental design. **b** IVIS imaging of tumor burden over time. **c** Quantification of tumor growth via luciferase intensity for mouse images in **b**. "PR" indicates partial response which is defined by a final tumor size over baseline but <$10^9$ at day 33 (relative light units, RLU). **d** Survival of treated mice over time. For **d** a Mantel-Cox log-rank test was performed with a Bonferroni correction for multiple comparisons and "*" denotes a significance of $p < 0.01667$ for three comparisons, $n = 5$ mice. Exact $p$-values are $p = 0.0135$ for adaptor only and $p = 0.0031$ for SNAP-CAR T cells only. Source data are available as a Source Data file.

simulations against experimental results. With the exception of Rituximab, model simulations using literature values alone, resulted in good recapitulations of experimental data (average SSE with literature values = 1.03) (Supplementary Table S3). Next, using SciPy, we minimized the sum of the squared error to optimize the kinetic parameters and better fit the model to the experimental data for each antibody pair[47]. During parameter estimation, the literature values were used as initial estimates and bounded within one order of

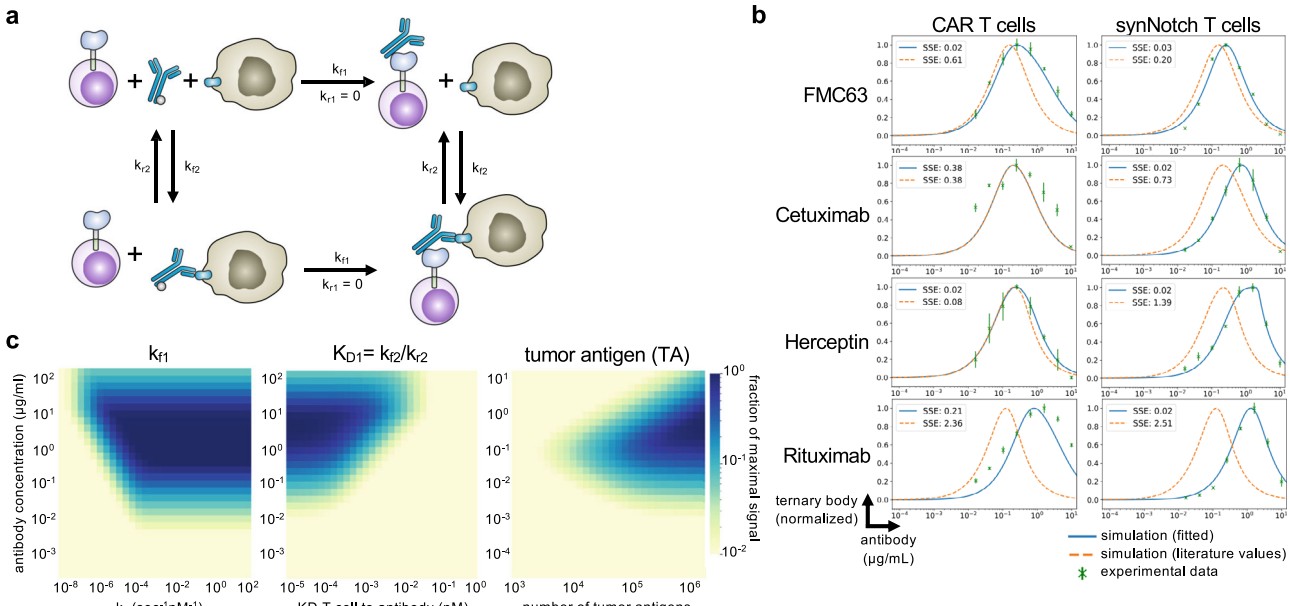

**Fig. 8 | Mathematical model of three-body binding in the context of antibody mediated T cell targeting. a** Schematic of the ODE model for SNAP receptor ternary body formation. **b** Model simulations using parameters from the literature and from parameter estimation, compared to experimental results for four different antibody and antigen pairs for SNAP-CAR and SNAP-synNotch receptors. **c** Parameter scans of $k_{f1}$ (binding rate of T cells to antibody), $K_{D1}$ (equilibrium dissociation constant between T cell and antibody), and the number of target antigens on the surface of the tumor. For experimental data in **c** (green), $n = 3$ biologically-independent experiments ± s.e.m. Source data are available as a Source Data file.

magnitude. With these constraints, we were able to minimize our model error in seven of the experimental results to (average SSE after fitting = 0.09) (Supplementary Table S3).

With a validated model, we next aimed to use the model to predict how different system parameters would affect receptor signaling and conducted parameter scans with the model. We first varied $k_{f1}$, the forward reaction rate of the antibody binding to the T cell receptor, to simulate the effects of increasing the on-rate, which could also be experimentally varied by changing the number of BG motifs conjugated to an antibody. We found that the model predicted that increasing the number of BGs per antibody would lead to greater ternary body formation at higher concentrations of antibodies. Once the $k_{f1}$ rate becomes greater than a threshold of $10^{-3}$ nm$^{-1}$sec$^{-1}$, this effect was expected to plateau. (Fig. 8c). Next, we scanned different values for the antibody to T cell affinity, the parameter maximized by the use of the covalent SNAPtag labeling. We found that stronger affinity was predicted to lead to ternary body formation over a wider range of antibody concentrations and lead to a higher overall level of ternary body formation. Finally, as target antigen concentration can vary based on the antigen being targeted, and for cancer, expression levels can also significantly vary greatly between patients and on cells within the same patient, we performed a parameter scan varying the level of tumor antigen expression. We found that greater antigen concentrations were expected to broaden the effective antibody dosage window for successful ternary complex formation, while lower antigen levels were predicted to require a higher amount of antibody to induce signaling and were predicted to be more susceptible to inhibition by the hook effect.

## Discussion

The universal adaptor SNAP-synNotch system further increases the versatility of the synNotch receptor framework leading to post-translational control of receptor signaling by controlling the co-administered antibody dose, as well as the ability to target multiple antigens using a single genetically encoded receptor. Our initial unsuccessful attempts to create an adaptor synNotch receptor using a

non-covalent interaction (between mSA2 and biotin) suggest that a very high-affinity interaction – such as a covalent bond – between the synNotch receptor and the tag is required for the adaptor system to function; presumably since the signaling mechanism for Notch is based on a pulling force[37]. The covalent bond produced by the SNAP enzyme and the benzylguanine moiety provides the tightest bond – a covalent bond – and thus maximizes this critical parameter. Indeed, the SNAP-synNotch receptor was able to be activated by an antibody adaptor when added in solution to the cell co-incubations, or when pre-loaded onto the target cells and was able to activate a therapeutically relevant cytokine response gene. While multiple adaptors were able to induce synNotch response gene activation, the kinetics of activation differed somewhat between antibody adaptors and were affected by adaptor dose as well as the target cell to synNotch cell ratio, indicating that some optimization will likely be needed for specific clinical indications. We were originally surprised that the pre-assembled SNAP-synNotch receptors were not functional and instead require pre-targeting of the cancer cells by the adaptor. However, this result can potentially be explained by again considering the mechanism of signaling, in which the receptor is proteolytically cleaved and thus destroyed following activation, not allowing for multiple signaling events from recycled receptors. The observed lack of response gene activation suggests that multiple bursts of receptor activation from distinct receptors over time may be needed to sufficiently trigger synNotch signaling. The universal SNAP system will open many opportunities for multi-antigen targeting using the synNotch receptor framework.

The SNAP-CAR displayed versatile and potent activity in experimental testing and has several beneficial characteristics over non-covalent adaptor CAR technologies[17–23]. In vitro testing in both Jurkat cells and primary human T cells showed efficient antigen targeting and activation of effector functions, including tumor cell lysis and cytokine production, with several different adaptors. Notably, the receptors were more reactive at lower antibody adaptor concentrations compared to our previously developed mSA2-CAR system (peak activity between 0.1–1 µg/mL for SNAP-CAR versus 25 µg/mL for the

non-covalent mSA2-CAR). Our results, supported by our modeling analysis, and the results of others, suggest that the affinity of the interaction between the CAR and the adaptor molecule is a key parameter for productive receptor signaling[17,20]. While many antibodies will be functional with a non-covalent, lower-affinity adaptor CAR, our model predicts that covalent bond formation could enable the use of antibodies, and other ligands, that otherwise have a binding affinity to the target antigen that is too weak to elicit an effect. The ability to create functional CARs by preloading the SNAP receptor, followed by removal of excess BG-antibody, provides unique opportunities to test candidate antigen binding regions as components of traditional CARs. Compared to CARs binding to adaptors through a transient interaction, the covalently assembled receptor will more closely resemble that of a traditional CAR. As the pre-assembled CARs were capable of signaling, pre-labeling the SNAP-CAR T cells could be a potential clinical approach, however, upon T cell activation, the cells would be induced to expand, thus diluting out the assembled receptor, requiring supplementation of additional antibody. The observed tumor clearance by pre-loaded SNAP-CAR T cells with additional adaptor infusions supports this potential therapeutic administration scheme. Additionally, the SNAPtag enzyme reacting with the bio-orthogonal benzylguanine grants the CAR exquisite specificity, and, being an enzyme of human origin, the SNAP protein is likely to be well-tolerated in a human host. This characteristic satisfies a key requirement for the persistence of the adoptively transferred therapeutic cells and minimizes the possibility of toxicities resulting from their immune rejection[48–50]. The SNAP-CAR's ability to perform simultaneous antigen targeting via combining multiple adaptors will allow for the potential to avoid cancer relapse through antigen loss, a hallmark of the adaptor CAR technologies. Finally, the inhibitory activity of the clinically relevant O6-BG molecule on SNAP-CAR activity could provide a convenient means to tune or turn off the activity of SNAP-CAR T cells as a safety measure without the need for a gene suicide switch[51].

The in vivo functionality of the SNAP-CAR system is a notable advance in the field of covalent enzyme therapeutics. The circulating adaptors showed efficient labeling of the SNAP-CAR T cells in the blood of mice with receptor assembly correlating with LNGFR marker expression. The SNAP-CAR T cells administered with an adaptor were also able to mediate potent anti-tumor activity comparable to that of a traditional CAR. While previous reports have utilized the SNAP/BG interaction for covalent cell labeling and imaging in vivo[52–54], our data further support its use for therapeutic applications.

Following the demonstration of activity in a pre-clinical animal model, future developments of the SNAP adaptor systems to clinical applications will be important in several key areas. Developing site-specific tagging approaches will help to lead to more homogeneous antibody-BG conjugates and potentially identify optimal BG-conjugation sites that could further maximize receptor signaling output. For any specific disease indication, there is the need to determine the optimal antibody or antibodies, to combine with the adaptor T cells to provide disease-specific or at least disease-associated induction of receptor signaling. Based on our results, additional self-labeling or covalent protein assembly systems could also provide good frameworks for universal adaptor CARs. Indeed recent work developing an adaptor CAR using the Spytag/SpyCatcher domains showed potent targeting capability[55]. Additional potential systems include candidates such as: CLIPtag, Halotag, SnoopTag, Isopeptag, Sortase, or split inteins[26,56–59]. It is possible that CARs made with other self-labeling enzymes, having different binding kinetics /dissociation rates, molecular size, and variation in expression level, could provide more robust signaling depending on the characteristics of the antigen(s) being targeted. Additionally, other synthetic receptor platforms may be amenable to the universal adaptor format by creating receptors with the SNAPtag protein domain[60,61].

Our molecular model of universal receptor systems provided key insights into our observed signaling behaviors and yielded predictions on how to potentially optimize receptor function. The model results suggested that the binding strength between the CAR and the adaptor is a critical parameter for signaling and that our SNAP receptors for which this interaction strength is maximized via a covalent bond are expected to be desirable. Furthermore, the model suggested that one way to improve activity would be to increase the forward reaction rate of the CAR binding to the adaptor, which could potentially be accomplished by increasing the number of BG molecules per antibody or by further mutations to the SNAP domain. Lastly, the model predicted that using adaptor CARs to target antigens that are expressed at high levels is preferable as these antigens would be expected to induce receptor signaling at lower antibody concentrations and would be less susceptible to the hook effect at higher antibody doses.

The field of universal receptor engineering is rapidly progressing, including innovations that enhance targeting specificity and adaptor versatility, as well as clinical testing in early-stage trials[24,62]. To enhance specificity, researchers have developed combinatorial antigen targeting approaches including "AND" logic-based targeting of cells with two antigens on their surface[17,63]. Additionally, emerging approaches include conditional spatial and temporal control of universal CAR T cells by stimuli such as small molecule drugs and UV light[64,65]. Further expanding the versatility of universal CARs, researchers have demonstrated targeting via adaptors constructed with molecules beyond IgG antibodies including nanobodies, DARPins, and small molecule drugs[55,66–68]. Outcomes from several ongoing and planned clinical trials of universal CARs in hematological and solid tumor settings will provide critical information for further development and technological refinement[62,69]. Finally, pairing the universal targeting capability of adaptor CARs with allogeneic cell approaches and gene editing promises to provide ideal off-the-shelf cell therapeutics[69].

SNAP-synNotch and SNAP-CAR T cells provide a powerful adaptor strategy for fully programmable targeting of engineered cells to multiple antigens using covalent chemistry. These systems have the potential for clinical application and biotechnological utility by providing researchers with the ability to rapidly screen CAR and synNotch antibody candidates and to rewire and activate cellular programs in response to highly specific antibody-antigen interactions.

## Methods
### Construction of viral expression vectors
pHR_PGK_antiCD19_synNotch_Gal4VP64 and pHR_Gal4UAS_tBFP_PGK_mCherry were gifts from Wendell Lim (Addgene plasmid# 79125; http://n2t.net/addgene:79125; RRID:Addgene_79125 and Addgene plasmid# 79130; http://n2t.net/addgene:79130; RRID:Addgene_79130, respectively). Sequences for all receptor coding regions and response constructs are listed in Supplementary Table S4. To generate pHR-PGK-SNAP-41BBζ, a DNA fragment encoding SNAP-41BBζ was codon-optimized, synthesized (Integrated DNA Technologies), and cloned into the pHR-PGK vector backbone using isothermal assembly. To generate pHR-PGK-SNAP-synNotch-Gal4VP64 and pHR-PGK-mSA2-synNotch-Gal4VP64, DNA encoding the SNAP or mSA2 coding region was codon-optimized and synthesized (Integrated DNA Technologies) and cloned in place of the anti-CD19scFv in plasmid pHR-PGK-antiCD19-synNotch-Gal4VP64 (Addgene# 79125) using isothermal assembly. To generate pHR-Gal4UAS-IL7-PGK-mCherry, a DNA fragment encoding IL-7 was codon-optimized, synthesized (Integrated DNA Technologies), and cloned in place of TagBFP in the pHR_Gal4UAS_tBFP_PGK_mCherry vector backbone using isothermal assembly. Lentivirus was generated using the above-described transfer vectors following methods described previously in detail[70]. The MSGV1 and RD114 retroviral plasmids were a gift from Dr. U. Kammula. The SNAP-CAR was cloned with T2A-LNGFR into the MSGV1 backbone via

isothermal assembly into the NcoI and NotI restriction enzyme sites and the retrovirus was produced following established methods[71].

## Production of BG-antibody conjugates

Rituximab (Rituxan, Genentech), Cetuximab (Erbitux, Eli Lily), Trastuzumab (Herceptin, Genentech) and FMC63 (Novus Biologicals) underwent buffer exchange into PBS using 2 mL 7 K MWCO Zeba Spin Desalting Columns (ThermoFisher Scientific). The Rituximab Fab fragment was generated using the Fab Preparation Kit (Pierce) following the manufacturer's protocol. Antibodies were then co-incubated with a 20-fold molar excess of BG-GLA-NHS (NEB) for 30 minutes at room temperature, followed by buffer exchange into PBS using 2 mL 7 K MWCO Zeba Spin Desalting Columns.

## Quantification of BGs on BG-conjugated antibodies

For in vitro conjugation of whole antibodies with SNAPtag, BG-conjugated purified antibodies (0.5 μg) were incubated with recombinant SNAPtag protein (2 μg). The solution was incubated in PBS (10 μL, pH 7.4) containing DTT (1 mM) at 37 °C for 2 h. Conjugation solutions were then diluted with Laemmli buffer, boiled for 5 minutes, and analyzed on an 8% SDS-PAGE (120 V, 1.5 h). Gels were visualized using imidazole-SDS-Zn reverse staining. Briefly, gels were stained with a 200 mM imidazole aqueous solution containing 0.1% SDS for 15 minutes with light agitation. The staining solution was decanted and replaced with water. After 30 seconds, the water was decanted and the gel was developed for 45 seconds with a 200 mM ZnSO4 aqueous solution with light agitation. The gel was then rinsed under running water for 10 seconds. Gels were imaged on a ChemiDoc with Image Lab Version 6.1.0 software (Bio-Rad) using epi white light on a black background. Relative band intensities were quantified with ImageJ. A correction factor of 1.5 was applied to the average number of BG/antibodies to account for the light chain. Light chains were conjugated to SNAPtag in the same manner except for 3 μg of antibody was incubated with 6 μg SNAPtag, gels were analyzed on a 10% SDS-PAGE (120 V, 1.2 h) and stained with Coomassie, and a correction factor was not applied.

## Cell line culture

Human tumor cell lines Jurkat Clone E6-1 (TIB-152), ZR-75-1(CRL-1500), K562 (CCL-243), SKOV-3(HTB-77), and Raji (CCL-86) were obtained from American Type Culture Collection (ATCC). NALM6 cells stably expressing firefly luciferase were a gift from G. Delgoffe. Cell lines were cultured at 37 °C in RPMI medium supplemented with 1X MEM amino acids solution, 10 mM Sodium Pyruvate, 10% fetal bovine serum (FBS), and Penicillin-Streptomycin (Life Technologies). K562 + EGFRt, K562 + CD20, NALM6 + CD20, NALM6 + HER2, and Jurkat+EGFRt cells that stably express full-length CD20, EGFRt https://www.ncbi.nlm.nih.gov/pmc/articles/PMC3152493/, and HER2, were generated by transducing cells with the indicated tumor antigen-expressing lentivirus and sorting for cells positive for antigen expression. To create the SNAP-CAR stable cell line, Jurkat cells were transduced with SNAP-41BBζ, and underwent fluorescence-activated cell sorting (FACS) for TagBFP expression and reporter (mCherry +) expression. To generate SNAP-synNotch lines, SNAP-synNotch-Gal4VP64 was co-transduced with either pHR-Gal4UAS-tBFP-PGKmCherry or pHR-Gal4UAS-IL7-PGKmCherry lentivirus, and receptor and response construct positive cells were obtained by FACS for anti-Myc-Tag antibody staining (Cell signaling Technology) and mCherry expression, respectively. HEK293T cells (ATTC, CRL-3216) and HEK293-GP cells (gift from U. Kammula, University of Pittsburgh), used for lentivirus and retrovirus production, respectively, were cultured at 37 °C in DMEM supplemented with 10% FBS and Penicillin-Streptomycin. Cell lines were authenticated for antigen expression by flow cytometry staining using indicated antibodies as shown in Supplementary Fig. S2. Cells were routinely tested for mycoplasma contamination, and all of the lines used tested negative for mycoplasma. No commonly misidentified cell lines were used except for HEK293T cells which were used for lentivirus production due to the ease of transfecting them with viral plasmids.

## Primary human T cell culture and transduction

All primary T cells for experiments were sourced from deidentified human Buffy Coat samples purchased from the Pittsburgh Central Blood Bank fulfilling the basic exempt criteria 45 CFR 46.101(b)(4) in accordance with the University of Pittsburgh IRB guidelines. PBMC were isolated from Buffy Coats from healthy volunteer donors using Ficoll gradient centrifugation, and human T cells were isolated using the Human Pan T cell isolation kit (Miltenyi Biotec). Human T cells were cultured in supplemented RPMI media as described for cell lines above, however, 10% Human AB serum (Gemini Bio-Products) was used instead of FBS, and the media was further supplemented with 100 U/ml human IL-2 IS (Miltenyi Biotec), 1 ng/ml IL-15 (Miltenyi Biotec), and 4mM L-Arginine (Sigma Aldrich). T cells were stimulated and expanded using TransAct Human T cell activation reagent (Miltenyi Biotec). For transduction, 48 h after activation, lentivirus was added to cells at a multiplicity of infection of 10–50 in the presence of 6 μg/ml of DEAE-dextran (Sigma Aldrich). After 18 h, cells were washed and resuspended in fresh T cell media containing 100 U/ml IL-2 and 1 ng/ml IL-15. Cells were split to a concentration of $1 \times 10^6$/mL and supplemented with fresh IL-2 and IL-15 every 2–3 days. After 10–12 days of stimulation and expansion, transduced cells were evaluated for CAR expression by flow cytometry and evaluated for activity in subsequent functional assays. CAR T cells for the mixed target lysis assay in Fig. 4e and cells made via gammaretroviral vectors were produced using a modified protocol. Briefly, isolated PBMCs we supplemented with 2 μg/ml OKT3 (Miltenyi Biotec) and 300 U/ml human IL-2 (Miltenyi Biotec) for 3 days and transduced by spinning cells on retronectin (RN) (Takara Bio) and virus coated plates. To coat the plates, 10 μg/mL RN in PBS was added to a plate at 4 °C for 24 h. RN was removed and 2-4 ml of viral supernatant was added to each well and spun at 2000 x *g* for 2 h at 32 °C. After removing 2 ml of media, $1 \times 10^6$ cells in 4 ml were added per well and spun at 32 °C for 10 mins at 1000 x *g*. For cells in Fig. 4e, $10^5$ T cells were sorted (FACS) for TagBFP expression. Sorted cells were further expanded by incubating with $3 \times 10^7$ irradiated PBMC feeder cells supplemented with 50 ng/mL OKT3 (Miltenyi Biotec) and 300 U/ml human IL-2 (Miltenyi Biotec) for 10 days and additional IL-2 supplementation every 2–3 days.

## Flow cytometry staining

Cells were washed and resuspended in flow cytometry buffer (PBS + 2% FBS) and then stained using the indicated antibodies for 30 minutes at 4 °C followed by two washes with flow cytometry buffer. Live cells and singlets were gated based on scatter. To evaluate SNAP-CAR and SNAP-synNotch expression, $1 \times 10^6$ cells were labeled with SNAP-Surface 647 1:2000 in complete cell media) for 30 minutes at 37 °C and washed three times in complete culture media. SNAP-synNotch Jurkat cells were additionally stained with anti-mycTag-AF488 antibody 1:50 (Cell Signaling Technology) to label the mycTag on the N-terminus of the receptor. To identify LNGFR + cells, and CD4 and CD8 positive T cells, the following antibodies were used: anti-CD271-BV421(BD Biosciences) 1:100, anti-CD4-BUV395(BD Biosciences) 1:100, anti-CD8-PECy7(BioLegend) 1:100. To identify human T cells loaded with antibody adaptors, cells were stained with either anti-human IgG-Fcgamma-AF647(Jackson ImmunoResearch) 1:200 or anti-human-IgG(H + L)-AF647(Jackson ImmunoResearch) 1:200. To identify human T cell subsets in murine blood, in addition to LNGFR, CD4 and CD8, we stained T cells with anti-CD62L-FITC (BD Biosciences) 1:100 and anti-CD45RA-BV785 (BioLegend) 1:100. To identify NALM6 + CD20 target in murine blood, cells were stained with anti-CD19-BV605 (BD Biosciences) 1:100 or PE

(BioLegend) 1:100 and anti-CD20-BV421 (BioLegend) 1:100. Validation provided by supplier e.g., antibodies from BioLegend and BD Bioscience have been widely used and their use is cited on their websites. Additionally, each antibody was tested on cells known to be negative and positive for the targeted antigen e.g. the anti-CD271(LNGFR) antibody was tested on un-transduced (mock) primary human T cells and CAR-transduced primary human T cell populations. All experiments were run on the LSRFortessa flow cytometer (BD Biosciences) and data were collected using the BD FacsDiva software v9.0 (BD Biosciences). Flow cytometry data was analyzed using FlowJo v10.8.1 (FlowJo, LLC), and data was presented and analyzed using Graphpad Prism v9 (GraphPad Software, LLC).

### SNAP-synNotch cell and target cell co-incubation assays for antibody-mediated activation

100,000 synNotch Jurkat effector cells were co-cultured with 200,000 of the indicated target cells and BG-conjugated antibody for 48 h. For assays with SNAP-synNotch cells engineered with the pHR_Gal4UAS_tBFP_PGK_mCherry response construct, co-incubated cells were evaluated by flow cytometry, gating for synNotch cells by mCherry positivity, and then quantifying TagBFP fluorescence for this mCherry + population. A representative plot summarizing the synNotch co-incubation gating scheme is shown in Supplementary Fig. S18. For assays with SNAP-synNotch cells engineered with the pHR_Gal4UAS_IL7_PGK_mCherry construct, following a 48 h co-incubation, cells were spun down and supernatants were collected and analyzed by ELISA for IL-7 following the manufacturer's recommended protocol (Peprotech). Absorbances were read using the SpectraMax i3 plate reader running the SoftMax Pro 7 software.

### SNAP-CAR T cell and target cell co-incubation assays for antibody-mediated activation

A total of 100,000 SNAP-CAR Jurkat or primary human T cell effector cells were co-incubated with 200,000 of the indicated target cells and antibody concentrations for 24 h and assayed by flow cytometry for T cell marker gene expression. For primary cell assays, cells were stained with anti-CD69-PE (BD Biosciences) 1:100 or anti-CD69-BV711(BD Biosciences) 1:100, anti-CD62L-FITC (BD Biosciences) 1:100, and anti-CD107a-APC (BD Biosciences) 1:100 antibodies and for Jurkat effector assays, cells were stained with anti-CD62L-FITC (BD Biosciences) 1:100 and anti-CD25-APC (BD Biosciences) 1:100 antibodies. For flow cytometry CAR + cells were analyzed by gating for the TagBFP+ population. Supernatants from primary cell assays were also collected and analyzed for IFNγ by ELISA (BioLegend). Absorbances were read using the SpectraMax i3 plate reader running the SoftMax Pro 7 software. All assays were performed in triplicate and average IFNγ production was plotted with standard deviation.

### Target cell lysis assay

The indicated target cells were stained with CellTrace Yellow following the manufacturer's recommended protocol (ThermoFisher), and 10,000 target cells per well were co-cultured with 50,000 SNAP-CAR T cells (E:T = 5:1) in a 96 well V-bottom plate with 1.0 μg/mL of the indicated BG-conjugated antibody. Plates underwent a quick-spin to collect cells at the bottom of the wells and were then incubated at 37 °C for 24 h. To identify lysed cells, co-incubated cells were stained with Ghost Dye Red Viability Dye (Tonbo Biosciences) and analyzed by flow cytometry. Target cells were identified by CellTrace Yellow and lysed target cells were identified by positive Ghost Dye staining. For the mixed target cell assay, Raji (CD20 + ) cells were stained with CellTrace Yellow. A total of 20,000 Raji, K562 + EGFRt, or Raji+K562-EGFRt (20,000 of each target) were co-incubated with 50,000 SNAP-CAR T cells and the indicated antibodies. K562 + EGFRt cells were identified by their membrane mCherry expression (EGFRt-[GGGGS]x3-mCherry). Percent

specific cytotoxicity of target cells was calculated by the equation: 100*(% experimental lysis – % target-only lysis) / (100 – % target-only lysis). A representative plot summarizing the CAR T cell and target cell co-incubation gating scheme is shown in Supplementary Fig. S19.

### Pre-labeling co-incubation assays

Pre-labeling co-incubation activation assays, were carried out as above, except prior to co-incubation for pre-labeled SNAP effector cell assays, SNAP-CAR or SNAP-synNotch Jurkat cells were first labeled with 1.0 μg/mL of the indicated BG-modified antibody in complete media for 30 minutes at 37 °C and then washed three times in complete media, and for pre-labeled target cell assays, target cells were labeled with 5.0 μg/mL of the indicated antibody for 30 minutes at 4 °C and washed two times with flow buffer. No additional antibody was added to these co-incubations. For primary human T cell pre-labeling studies, 2 to 20 million SNAP-CAR T cells were incubated with indicated dose of antibody adaptor for 1 h at 37 °C and washed. For animal studies a pre-labeling dose of 150 μg/mL Herceptin-BG adaptor was used, and following pre-incubation, cells were washed and resuspended in PBS at $20 \times 10^6/100\,\mu l$ per mouse.

### Mouse studies

Animal work in this study was approved by the University of Pittsburgh Institutional Animal Care and Use Committee (IACUC), and procedures were performed under their guidelines. Four to six-week-old female, NOD-SCID-γchain-deficient (NSG) mice (Jackson Laboratories) were used for all mouse experiments in the manuscript. For in vivo loading of SNAP-CAR T cells with antibody adaptor experiments, mice were injected with $10 \times 10^6$ SNAP-CAR T cells r.o. and 50 μg Herceptin-BG adaptor with or without 10 mg of human IgG antibody (GammaGard; Takeda Pharmaceuticals) were injected i.p. 24 h later, blood was drawn via submandibular vein bleeding, purified via anti-coagulation with Alsever's solution, RBC lysis, and washing, and analyzed by flow cytometry. For human tumor xenograft mouse model experiments, mice were injected with $0.5 \times 10^6$ NALM6 + HER2-luciferase tumor cells i.v. After four days mice were injected with $20 \times 10^6$ unsorted CAR T cells pre-loaded with or without 150 μg Herceptin-BG adaptor, and/or IVIG 10 mg i.p., $20 \times 10^6$ anti-HER2 CART cells i.v., or 150 μg Herceptin-BG adaptor alone i.p. IVIG and Herceptin-BG adaptor injections were repeated at a 3-day interval. Tumor burden was measured every 5 days by injecting D-luciferin (GoldBio), 33.3 μg/mouse, and incubated for 9 minutes. Luminescence in mice was acquired and quantified using the IVIS Lumina XR imaging platform (Perkin Elmer) using the Living Image v4.3.1 software (Caliper LifeSciences). For CD20-targeted SNAP-CAR experiment, methods were performed as above for HER2 except that CAR T cells were injected three days after tumor injection instead of four. At day 35 for the HER2 experiment and day 40 for CD20 experiment, blood was drawn via submandibular vein bleeding, purified via anti-coagulation with Alsever's solution, RBC lysis, and washing, and stained with indicated antibodies and analyzed by flow cytometry. For the CD20 targeting experiment, splenocytes were harvested and analyzed, briefly: spleens were extracted into RPMI + 10%FBS, mashed and filtered through a 70 μM filter, washed in FACS buffer, underwent RBC lysis and washing, stained with indicated antibodies, and analyzed by flow cytometry.

### Mathematical model

The model for ternary body formation considered the following 8 binding reactions between the tumor cells, T cells, and antibody with six different species: T cell (*Tc*), antibody (*Ab*), tumor cell (*Tu*), T cell bound to antibody (*Tc.Ab*), tumor cell bound to antibody (*Ab.Tu*), and a ternary body complex of a T cell bound to antibody and tumor cell (*Tc.Ab.Tu*) and where rates $k_{fi}$ (*i* = 1.4)

represent the forward kinetic rate constants, and rates $k_{ri}$ represent the reverse kinetic rate constants:

$$\text{Reactions 1 and 2}: \ Tc + Ab \underset{k_{r1}}{\overset{k_{f1}}{\leftrightharpoons}} Tc.Ab$$

$$\text{Reactions 3 and 4}: \ Ab + Tu \underset{k_{r2}}{\overset{k_{f2}}{\leftrightharpoons}} Ab.Tu$$

$$\text{Reactions 5 and 6}: \ Tc.Ab + Tu \underset{k_{r3}}{\overset{k_{f3}}{\leftrightharpoons}} Tc.Ab.Tu$$

$$\text{Reactions 7 and 8}: \ Tc + Ab.Tu \underset{k_{r4}}{\overset{k_{f4}}{\leftrightharpoons}} Tc.Ab.Tu$$

From reactions 1-8, we derived a system of equations to describe the accumulation of each of the six species in the model. In Eqs. 1–8 below, we list the forward and backward components of the eight reactions expressing the change in concentration of each species:

$$rxn_1 = k_{f1}*[Tc]*[Ab] \text{(binding of T cell to antibody)} \quad (1)$$

$$rxn_2 = k_{r1} \cdot [Tc.Ab] \text{(dissociation of T cell} - \text{antibody)} \quad (2)$$

$$rxn_3 = k_{f2} \cdot [Ab]*[Tu] \text{(binding of tumor cell to antibody)} \quad (3)$$

$$rxn_4 = k_{r2} \cdot [Ab.Tu] \text{(dissociation of tumor cell} - \text{antibody)} \quad (4)$$

$$rxn_5 = k_{f3} \cdot [Tc.Ab] * [Tu] \text{(binding of T cell} - \text{antibody to tumor cell)} \quad (5)$$

$$rxn_6 = k_{r3} \cdot [Tc.Ab.Tu] \text{(dissociation of tumor cell from ternary body)} \quad (6)$$

$$rxn_7 = k_{f4} \cdot [Tc] \cdot [Ab.Tu] \text{(binding of T cell to tumor cell} - \text{antibody)} \quad (7)$$

$$rxn_8 = k_{r4}*[Tc \cdot Ab \cdot Tu] \text{(dissociation of T cell from ternary body)} \quad (8)$$

The Eqs. 9–14 below were used to compute the change in concentration of each species.

$$\frac{d[Tc]}{dt} = -rxn_1 + rxn_2 - rxn_7 + rxn_8 \text{(change in free T cell receptor)} \quad (9)$$

$$\frac{d[Ab]}{dt} = -rxn_1 + rxn_2 - rxn_3 + rxn_4 \text{(change in free antibody)} \quad (10)$$

$$\frac{d[Tu]}{dt} = -rxn_3 + rxn_4 - rxn_5 + rxn_6 \text{(change in free tumor cell receptor)} \quad (11)$$

$$\frac{d[Tc.Ab]}{dt} = +rxn_1 - rxn_2 - rxn_5 + rxn_6 \text{(change in T cell} - \text{antibody)} \quad (12)$$

$$\frac{d[Ab.Tu]}{dt} = +rxn_3 - rxn_4 - rxn_7 + rxn_8 \text{(change in tumor cell} - \text{antibody)} \quad (13)$$

$$\frac{d[Tc.Ab.Tu]}{dt} = +rxn_5 - rxn_6 + rxn_7 - rxn_8 \text{(change in ternary body)} \quad (14)$$

The ODE model was created under the assumption that the system components were well-mixed. Variables used in the ODEs were taken from the experimental design and literature values of kinetic binding and dissociation rates as summarized in Supplementary Table S2[43–46]. The ODE model was written in Python3.7 and solved using SciPy1.7.3. Code is publicly available: https://github.com/pitt-miskov-zivanov-lab/TernaryBody. To examine the concentration of each species with time, the system of ODEs was solved using the initial conditions and experimental setup values through a kinetic simulation (Supplementary Fig. S20). To generate equilibrium simulations (Supplementary Fig. S21), kinetic simulations were run for variety of antibody concentrations ($10^{-4}\,\mu g/mL$ – $10^1\,\mu g/mL$) and total ternary body formation from the equilibrium state of each kinetic simulation was plotted. To fit the model, we calculated the sum of squared error (SSE) between the experimental data and the simulation results. For the experimental data we used the TagBFP MFI for synNotch (Fig. 2d) and CD25 MFI for the read-out of SNAP-CAR activation (Fig. 3d). As the experimental data was only collected at specific points of antibody concentration, only the matching points in the simulations were used. Using SciPy, we minimized SSE to optimize the kinetic parameters and better fit the model to the experimental data for each antibody pair (Fig. 8b). During parameter estimation, the literature values were used as initial estimates and bounded within one order of magnitude. Parameter scans of $k_{f1}$, $K_{d2}$, and the number of tumor antigens were conducted as above for equilibrium simulations using 900 simulations over the bounds for each parameter. Ternary body formation was normalized to the maximal concentration across all simulations. Supplementary Table S5 contains a list of fitted parameter values.

## Statistical methods and data analysis

The number of replicates, mean value, and error are described in the respective figure legends and/or methods. Error bars are shown for all data points with replicates as a measure of variation within a group. Statistical methods were not used to predetermine the sample size. Sample sizes for experiments were determined from analogous studies performed by us and others where the differences between groups were expected to be comparable, and the same statistical methods could be applied. A minimum of triplicates was chosen to allow for the calculation of statistics. For flow cytometry, >10,000 events were collected to characterize a distribution of the data. Biologically independent experimental triplicates then allowed for statistical analysis of data features. No data were excluded. All key experiments were performed at least twice. Results were consistent across these replicates and the data presented in the article is representative of the trends we observed. Samples were allocated to identical wells spatially in a manner that facilitated experimental organization. There is no reason to believe the spatial location of the sample influenced experimental results. For in vivo mouse experiments, after tumor injection, but before experimental treatment, mice were evaluated for tumor growth and evenly

distributed for luminescence with each group having some larger and some smaller tumor sizes. This re-distribution was performed to ensure that post-therapy tumor growth was not affected by the tumor size prior to therapeutic treatment as larger initial tumors could be expected to grow more rapidly. Blinding was not performed; however, the authors agree that samples were processed uniformly when acquiring data regardless of whether they were controls or experimental samples. As the data presented is quantitative in nature, blinding was unnecessary for the experiments performed.

### Reporting summary
Further information on research design is available in the Nature Portfolio Reporting Summary linked to this article.

## Data availability
All data generated in this study are provided in the Supplementary Information/Source Data file. Source data are provided with this paper.

## Code availability
Code for the ODE mathematical model is publicly available: https://github.com/pitt-miskov-zivanov-lab/TernaryBody.

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

## Acknowledgements

This work was supported by NIH grant R01 GM142007 (J.L., A.D.); NIH grant R35 CA210039 (O.J.F.); NIH grant R21 AI130815 (A.D.); DARPA award W911NF-17-1-0135 (N.M-Z.); AIRC postdoctoral fellowship 22321 (E.R.); and by the Michael G. Wells Prize (J.L.). This work benefitted from using the SPECIAL BD LSRFORTESSA funded by NIH 1S10OD011925-01. This project also used the Hillman Animal Facility, In Vivo Imaging Facility, and Cytometry Facility that are supported in part by award P30CA047904. Parts of Figs. 1, 2a, 3a, 5a, 5b, 6a, 6b, 8a, S5a and S15a were drawn by using pictures from Servier Medical Art. Servier Medical Art by Servier is licensed under a Creative Commons Attribution 3.0 Unported License (https://creativecommons.org/licenses/by/3.0/). Parts of Figs. 1c, 2a, 6b, 7a, S5a and S15a were created with BioRender.com.

## Author contributions

J.L. and A.D. designed the research. A.A.B and N.M-Z. created the mathematical model, and A.A.B. performed the model data fitting and parameter scans. E.R., J.L., Y.T., V.S., A.P., E.L.A and M.K. carried out the experiments and analyzed the data. J.L., N.M-Z., A.D., and O.J.F. supervised the work. E.R., J.L., A.A.B., Y.T., A.P., M.K., N.M-Z., A.D., and O.J.F. interpreted the data. J.L., E.R., and A.D. and wrote the manuscript with input from all authors.

## Competing interests

J.L. and A.D. are inventors on a patent application filed by the University of Pittsburgh on the universal SNAP receptor technology described herein (WO2020072764A1). The remaining authors declare no competing interests.
