## [Peer Review File · Nature Communications]

Reviewers' Comments:

Reviewer #1:

Remarks to the Author:

In this manuscript, Lohmueller et al developed a "switchable" SNAP-synNotch receptor and SNAP-CAR systems that could be used with benzylguanine (BG)-conjugated antibodies to target cancers. In addition, the authors developed a mathematical model of switchable receptor systems and predicted cancers with high expression of tumor-antigens could be targeted using this system.

The use of the synNotch concept to develop a CAR system is novel. This system allows using one CAR to target tumors with different antigens, which is highly desirable. In current CAR T cell field, most of the systems can only target one antigen, and thus, strategies targeting different target antigens are highly desirable. There are studies using dual or more CARs. However, these systems are expensive, and do not allow the flexibility.

This manuscript is overall well written and there are a number of novel aspects, including the development of a novel SynNotch CAR and the mathematical model. However, there are a few key issues need to be addressed:

1. The in vitro responses are excellent, but there is no in vivo data provided. The trafficking of T cells to the tumors and the persistence of the CAR T cells is already a complicated process. In a more complicated system like this, the binding of antigen, antibody and receptors in vivo could be very difficult to predict. There should be data demonstrating that this system have some in vivo effect. In addition, the effect should be compared with a classic CAR.
2. Although this system provides flexibility for targeting cancers with different antigens, there is no data demonstrating this flexibility—i.e. a mixture of cancer cells with different antigen expressions.
3. One of the advantage of the syn-Notch system is the capability to modify the output signals. However, in the syn-CAR system, the authors did not demonstrate this. It was shown in Figure 2 Jurkat cells produced IL-7. Is this replicable using the syn-CAR? If this is not possible, what is the difference between this syn-CAR, and a normal CAR binding to certain part of antibodies?

Minor points:

1. The Notch system should be explained with more details.
2. The results section starts from the unsuccessful story of mSA2 binding to the biotinylated antibodies. It gives the story a negative tune. Will the authors consider moving this section to a later part of the article? This is an important finding that defines the binding threshold for a functional synNotch. Maybe if this section is moved to a later part, the article will flow better.
3. Figure 2a and 2b need more explanation, i.e. Gal4-VP64 is not explained.
4. The y-axis of 2e should be IL-7, instead of IL-17.
5. The kinetics of the antibody activation of the cells vary a lot in 2D. There should be some explanation/discussion on this. Does it mean the system needs optimization for each antibody? This will be very difficult to use in treatment settings.
6. In Figure 3, is it possible to add BFP data along with CD25 and CD62L?
7. In Figure 4C, are the cells gated on Snap-CAR+ cells? What is the CD4 vs CD8 ratio for the transduction? What are the memory phenotypes of the transduced cells? Suggest to add the information to supplementary.
8. In Figure 4, when using Fab Rituximab, was the Fab labelled with the same number of BG to the full-length antibody?
9. In Figure 5, it is not clear whether the effector cells are Jurkat cells or the human T cells. Human T cells should be used here and cell lysis assay should be performed.
10. In Figure 6b, Cetuximab CAR T cell panel, the blue line is missing.
11. Ref 38 on page 27's format is wrong.
12. Figure 4 legend, CD105a should be CD107a.
13. Figure S2C, +2 SNAPtag label should be moved down.
14. In Figure S2, why are the full-length antibodies only 55KDa?
15. In Figure S4, is it possible to change the table to curves? It is difficult to visualize the data. If it is not changed, please indicate the color intensity of the heatmap.
16. In Figure S5, T cell.Antibody line is missing.

Reviewer #2:

Remarks to the Author:

To expand the toolkit of parts available for modular receptor engineering, the authors engineered a "switchable" adaptor receptor system using post-translational, covalent attachment of an antibody to the extracellular domain of engineered receptors via SNAPtag technology. The novel aspects of this work include the use of a covalent labeling strategy with CARs and the extension of this adaptor approach to synNotch receptors, both of which would be of use to the field. While other adaptor-based CAR systems have been reported, this is the first adaptor synNotch.

However, there are several concerns that should be addressed:

1. Although the use of a covalent association strategy is a new approach to build adaptor receptor systems that have previously been constructed with weaker non-covalent interactions, the marginal impact of this approach is not entirely clear. Though an experimental comparison between the SNAPtag-based system and other existing adaptor systems is likely unnecessary, a model-based investigation could help to elucidate the regime in which covalent linkage would be advantageous. For example, the authors comment in the discussion that a scenario in which the antibody-target affinity is weaker could require a covalent adaptor system, but these concepts are only treated qualitatively. More quantitative analyses would help to define the impact of the SNAPtag platform on performance.
2. The use of the word "switchable" to describe this system is potentially misleading; "switchable" is often used to describe the ability of a system to move between 'on' and 'off' states, but that quality does not exist in this system. Additionally, this word can be taken to indicate general reversibility, which is also not necessarily possible given the covalent bond. Alternatively, consider using the word "modular" or "universal".
3. In general, it would be useful to clearly state in the figure captions if sorting or gating strategies were used to produce the populations shown in figure panels. Some examples where this information is needed include Figure 2d-e and Figure 4b-d.
4. Page 4, paragraph 2: It would be useful to define the phrase "self-labeling".
5. Page 5, Results paragraph 1: references to Supplementary Figure 1 B & C are directed to the wrong panels within the figure and a reference to panel D is missing. Additionally, the stated K_d for msA1-biotin at the end of this paragraph seems erroneous; it should probably read 5.5×10^{-9} as opposed to 10^9 M.
6. Page 6, paragraph 1: I believe that I understand what is meant by the hook effect (when antibody is in excess, saturation of target sites by free antibody competes with receptors for target engagement), but it would be worth clarifying for the reader.
7. Page 9, paragraph 1: it could be useful to state in the main text the technique used to quantify target cell lysis.
8. Page 11, Figure 6b: can you speculate as to why the Cetuximab+CAR T cells case was not improvable through parameter fitting?
9. Page 12, Discussion paragraph 1: "synNoch" should be "synNotch".
10. Page 10 in reference to the experiments in Figure 5a-b. It would be useful to comment briefly, at this point, as to whether the observed patterns were unexpected and perhaps to briefly speculate as to why patterns may differ between synNotch and CAR across these panels.
11. Figure 2d-e: Y-axes are incorrectly labeled IL-17 and should be labeled IL-7.
12. Methods, Flow Cytometry Staining section: Was blocking of any sort used in surface staining procedures? Please describe if so.

13. Figure 5b: Data points on the right (Raji) blue bar of the SNAP-synNotch panel are not scattered in the same way that the others are.

14. Figure 6b: It would be useful to indicate how these models were fit (i.e., Were the synNotch and CAR models corresponding to a single antibody fit simultaneously?). Additionally, please report all fitted parameters.

Response to Reviewers

We thank the reviewers for their positive comments and suggestions which have directed us to substantially improve the manuscript. We have addressed each of their remarks in the manuscript text as described below in **bold**.

Reviewer #1 (remarks to the author):

1. *The in vitro responses are excellent, but there is no in vivo data provided. The trafficking of T cells to the tumors and the persistence of the CAR T cells is already a complicated process. In a more complicated system like this, the binding of antigen, antibody and receptors in vivo could be very difficult to predict. There should be data demonstrating that this system have some in vivo effect. In addition, the effect should be compared with a classic CAR.*

Answer: We agree with the reviewer about the importance of the *in vivo* data and now have now added the suggested *in vivo* mouse experiments to the manuscript. These new data include *in vivo* labeling of the SNAP-CAR T cells with to adaptor (Fig. 6) and human tumor xenograft experiments demonstrating potent and specific anti-tumor activity of the SNAP-CAR T cells *in vivo* (Fig. 7 & Supplementary Fig. S11). This data includes a direct comparison to a classic CART targeting the same antigen (HER2).

2. *Although this system provides flexibility for targeting cancers with different antigens, there is no data demonstrating this flexibility—i.e. a mixture of cancer cells with different antigen expressions.*

Answer: We have now added data showing the flexibility of the system to target a mixture of tumor cells with different antigen expressions (Fig. 4E). SNAP CAR T cells specifically lysed antigen+ tumor cells and adaptor combinations were able to elicit dual antigen targeting.

3. *One of the advantages of the syn-Notch system is the capability to modify the output signals. However, in the syn-CAR system, the authors did not demonstrate this. It was shown in Figure 2 Jurkat cells produced IL-7. Is this replicable using the syn-CAR? If this is not possible, what is the difference between this syn-CAR, and a normal CAR binding to certain part of antibodies?*

Answer: We have revised the text to further clarify the synNotch system and its differences with the SNAP-CAR system. Briefly, the synNotch receptor has the sole output of producing a transgene while the SNAP-CAR activates T cell signaling receptor pathway. The production of a transgene in response to CAR activation in addition to activation of TCR signaling is possible by delivery of a NFAT promoter-driven construct which has been demonstrated by several other groups. The unique aspect of the synNotch system is the ability to deliver the desired transgene without activating other pathways.

Minor points:

1. *The Notch system should be explained with more details.*

Answer: We have now added further explanation and description of the synNotch system in the Introduction and Results sections.

2. *The results section starts from the unsuccessful story of mSA2 binding to the biotinylated antibodies. It gives the story a negative tune. Will the authors consider moving this section to a later part of the article? This is an important finding that defines the binding threshold for a functional synNotch. Maybe if this section is moved to a later part, the article will flow better.*

Answer: We agree with this comment and have now edited the text to shorten this section and move it to a later place in the paper.

3. *Figure 2a and 2b need more explanation, i.e. Gal4-VP64 is not explained.*

Answer: We have now added further explanation of the synNotch system which includes a description of the Gal4-VP64 part.

4. *The y-axis of 2e should be IL-7, instead of IL-17.*

Answer: We have corrected this in the figure.

5. *The kinetics of the antibody activation of the cells vary a lot in 2D. There should be some explanation/discussion on this. Does it mean the system needs optimization for each antibody? This will be very difficult to use in treatment settings.*

Answer: We have added a discussion of this point into the Results and Discussion sections. In brief, the reviewer is correct that the curves are somewhat variable, and the system would likely require optimization in the clinic for a given adaptor. However, the qualitative behavior is still comparable showing universality and overall dosages that obtain peak activity are relatively similar across antibody/antigen pairs. It is likely that any approach that is being translated to the clinic will likely require dedicated focus and optimization.

6. In Figure 3, is it possible to add BFP data along with CD25 and CD62L?

Answer: To clarify, while BFP is the output gene for the synNotch system in Fig 2., in Fig. 3 the BFP is a marker gene for CAR expression. For the activation marker analysis, CAR T cells in the co-incubation assay are identified by the BFP marker gene with activation genes CD62L and CD25 being quantified for these BFP+ cells. We updated the legend in the Fig. 3. to better explain the analysis.

7. In Figure 4C, are the cells gated on SNAP-CAR+ cells? What is the CD4 vs CD8 ratio for the transduction? What are the memory phenotypes of the transduced cells? Suggest to add the information to supplementary.

Answer: Yes, the cells in Fig. 4C are gated on the SNAP-CAR+ population (BFP+). We have now included in the CD4 and CD8 flow cytometry data for the SNAP CAR T cells as well as the level of CD62L for these cells which informs on the effector/memory cell phenotype (Supplementary Fig. S9). These specific plots were characterizing the CAR T cells administered to mice (Fig. 7).

8. In Figure 4, when using Fab Rituximab, was the Fab labelled with the same number of BG to the full-length antibody?

Answer: The average number of BG molecules were quantified and listed in Table#S1. The average number of BG molecules per full-length Rituximab antibody (2.8) was similar to the Fab (2.5).

9. In Figure 5, it is not clear whether the effector cells are Jurkat cells or the human T cells. Human T cells should be used here and cell lysis assay should be performed.

Answer: The results in Fig. 5. A, B were performed using Jurkat effector cells, however we have now added the requested cell lysis assays using primary human CAR T cells (Fig. 5C) as well as T cell activation data with primary human CAR T cells (Supplementary Figs. S6 & S7).

10. In Figure 6b, Cetuximab CAR T cell panel, the blue line is missing.

Answer: In this figure the blue line is actually aligned with the orange curve so it is tricky to see, but it appears as a blue and orange dashed line.

11. Ref 38 on page 27's format is wrong.

Answer: We have corrected the formatting for this reference.

12. Figure 4 legend, CD105a should be CD107a.

Answer: We have corrected the figure legend.

13. Figure S2C, +2 SNAPtag label should be moved down.

Answer: Due to the non-traditional covalent bond leading to branched protein chains the proteins run at a slightly higher than expected length in the denaturing gel, so the +2 SNAPtag label is at the correct size for what we have observed a single antibody chain and two SNAP proteins.

14. In Figure S2, why are the full-length antibodies only 55KDa?

Answer: The SDS-PAGE gels are denaturing gels, so one full length antibody molecule will be dissociated into 2 heavy chains of ~50kDa each and 2 light chains of ~25kDa each.

15. In Figure S4, is it possible to change the table to curves? It is difficult to visualize the data. If it is not changed, please indicate the color intensity of the heatmap.

Answer: We have added a color key (now Supplementary Fig. S3) to make the heatmap easier to visualize and interpret.

16. In Figure S5, T cell.Antibody line is missing.

Answer: This line is tricky to see, but it is there on the plot very close to the brown dashed line for the "T cell.Antibody.Tumor" population.

Reviewer #2 (Remarks to the Author):

1. Although the use of a covalent association strategy is a new approach to build adaptor receptor systems that have previously been constructed with weaker non-covalent interactions, the marginal impact of this approach is not entirely clear. Though an experimental comparison between the SNAPtag-based system and other existing adaptor systems is likely unnecessary, a model-based investigation could help to elucidate the regime in which covalent linkage would be advantageous. For example, the authors comment in the discussion that a scenario in which the antibody-target affinity is weaker could require a covalent adaptor system, but these concepts are only treated qualitatively. More quantitative analyses would help to define the impact of the SNAPtag platform on performance.

Answer: Based on our results comparing the mSA2 synNotch receptor with the SNAPtag receptor, a very high affinity (subnanomolar at least) attachment is likely required for the creation of an adaptor synNotch system (likely due to its unique receptor signaling mechanism of receptor pulling and proteolytic cleavage vs. receptor clustering for a CAR) making this covalent interaction paramount. For the adaptor CARs, we now include a lengthy discussion of the advantages of the covalent interaction in the discussion section. While unfortunately it is very difficult to comment on the exact affinity range that of adaptors in precise quantitative terms, we do see higher peak activation of the SNAP-CAR at lower adaptor doses compared to our previously published mSA2 adaptor CAR system. We now also mention this comparison in the text.

2. The use of the word “switchable” to describe this system is potentially misleading; “switchable” is often used to describe the ability of a system to move between ‘on’ and ‘off’ states, but that quality does not exist in this system. Additionally, this word can be taken to indicate general reversibility, which is also not necessarily possible given the covalent bond. Alternatively, consider using the word “modular” or “universal”.

Answer: We have removed the term “switchable” and have instead used “universal” or “adaptor” to describe the system.

3. In general, it would be useful to clearly state in the figure captions if sorting or gating strategies were used to produce the populations shown in figure panels. Some examples where this information is needed include Figure 2d-e and Figure 4b-d.

Answer: We have now included text to each legend to better describe the gating strategies used.

4. Page 4, paragraph 2: It would be useful to define the phrase “self-labeling”.

Answer: We have now added text to define the term “self-labeling”.

5. Page 5, Results paragraph 1: references to Supplementary Figure 1 B & C are directed to the wrong panels within the figure and a reference to panel D is missing. Additionally, the stated K_d for mSA1-biotin at the end of this paragraph seems erroneous; it should probably read 5.5×10^{-9} as opposed to 10^{-9} M.

Answer: We have now corrected these errors (Note: Supplementary Fig. 1 is now Supplementary Fig. 4).

6. Page 6, paragraph 1: I believe that I understand what is meant by the hook effect (when antibody is in excess, saturation of target sites by free antibody competes with receptors for target engagement), but it would be worth clarifying for the reader.

Answer: We have added a description of the hook effect to the text to be clarify this phenomenon.

7. Page 9, paragraph 1: it could be useful to state in the main text the technique used to quantify target cell lysis.

Answer: We have now added a description to the main text to better describe this assay.

8. Page 11, Figure 6b: can you speculate as to why the Cetuximab+CAR T cells case was not improvable through parameter fitting?

Answer: During parameter estimation, the literature values were used as initial estimates and bounded to one order of magnitude. With these constraints, we were able to minimize model error for seven of the antibody/receptor combinations. The eighth case (SNAP-CAR T cells with Cetuximab) was not able to converge which was potentially a result of the kinetic parameters being too far from the optimal values. We speculate that it is possible that the antibody binding to EGFR antigen was more affected by the presence of the randomly conjugated BG chemical motifs than the other antibodies thus leading to altered kinetics for the antibody adaptor vs. the un-modified Cetuximab antibody.

9. Page 12, Discussion paragraph 1: “synNoch” should be “synNotch”.

Answer: We have corrected this typo.

10. Page 10 in reference to the experiments in Figure 5a-b. It would be useful to comment briefly, at this point, as to whether the observed patterns were unexpected and perhaps to briefly speculate as to why patterns may differ between synNotch and CAR across these panels.

Answer: We have included a discussion of the results including speculation on the differences of the synNotch and CAR constructs in the Results section.

11. Figure 2d-e: Y-axes are incorrectly labeled IL-17 and should be labeled IL-7.

Answer: We have corrected this error.

12. Methods, Flow Cytometry Staining section: Was blocking of any sort used in surface staining procedures? Please describe if so.

Answer: The FACS staining media contains 2% FBS which is noted, and no additional blocking is performed.

13. Figure 5b: Data points on the right (Raji) blue bar of the SNAP-synNotch panel are not scattered in the same way that the others are.

Answer: This observed difference in data point scattering has to do with auto-formatting of how the GraphPad Prism software plots the data. For data points that are potentially overlapping, the software plots them horizontally, while data points that are not overlapping are plotted vertically. The Raji data points of the SNAP-synNotch panel have a higher level of variation and thus are plotted vertically by the software.

14. Figure 6b: It would be useful to indicate how these models were fit (i.e., Were the synNotch and CAR models corresponding to a single antibody fit simultaneously?). Additionally, please report all fitted parameters.

Answer: The Methods section includes a description of the fitting process. Of note the synNotch and CAR models were fit separately, using the TagBFP MFI for synNotch (Fig. 2D) and CD25 MFI for the read-out of SNAP-CAR activation (Fig. 3D). We have now also included a table of the fitted parameter values (Table S5.)

Reviewers' Comments:

Reviewer #1:

Remarks to the Author:

Ruffo et al described the construction of two systems: 1. A synNotch receptor that a SNAPtag protein. The system was tested in Jurkat cells. 2. A CAR that was constructed by using a SNAPtag protein domain instead of a traditional scFv. The Snap-Tag CAR system was tested using transduced T cells from human PBMCs. The anti-cancer effect of the SNAPtag-CAR T cells was demonstrated in NSG mice bearing Nalm6+ Her2 mice.

The manuscript was well written, and the experimental design was clearly explained. It is a well-structured manuscript. Although SynNotch-CAR concept and Antibody-based universal CAR concepts have both been reported (i.e. Choe 2021 Sci Transl Med, PMID: 33910979; Kuo et al, JITC , PMID: 35728874), using the SNAPtag binding to BG labelled antibody design for CAR T cell mediated killing is novel.

There are a few points the authors may consider to address:

Major points:

1. The authors failed to demonstrate the SNAP-CARs being superior to the traditional scFv CAR. Most of the experiments were mainly carried out in vitro. The only in vivo efficacy experiment was shown in Figure 7. In this experiment, Nalm6 leukemia cells expressing Her2 injected i.v. in NSG mice was used as a model, and an anti-Her2 antibody was used as the adaptor antibody. The experiment should include an anti-CD19 antibody and compare with a CD19 CAR. The advantage of the universal CAR is the flexibility of multiple antigen targeting, but the authors did not demonstrate this in vivo.
2. If the authors claim an effect in solid tumors, the cancer model used in Figure 7 should be a solid tumor model instead of a leukemia cell line bearing a solid cancer antigen.
3. There is no characterization of the SNAP-CAR nor SynNotch T cells in vivo. For example, T cell proliferation, infiltration to the tumor sites and T cell persistence. Is the SynNotch/SNAP-CAR system better than the scFv-CARs, as the binding/signaling is different, or are they the same in these aspects?
4. For the cell culture protocols, the authors need to demonstrate their cellular products phenotypes, i.e. Tcm vs Teff phenotypes, and compare these with the traditional scFv-CAR T cell products, such as the Her2-CAR used in this manuscript.
5. Please include other work carried out in this field in the discussion section.

Minor points:

1. It's not clear when mentioned "Snap-synNotch cells" in the manuscript in multiple places. i.e. in figure 5, did the authors use PBMC transduced cells or Jurkat cells?

Reviewer #2:

Remarks to the Author:

The authors provided a thorough and thoughtful response to points raised in the first round of review. The added experiments, clarification, and minor reorganization included in this revision support key conclusions and strengthen the narrative arc. This revised article is a useful and interesting contribution to the field of synthetic receptor engineering.

The following optional suggestions are offered for consideration:

1. Figure 2d: It could be useful to publish single cell-based flow cytometry data of reporter expression as histograms for experiments presented in Figure 2d and Figure S3. This information could elucidate how the proposed strategies to tune receptor signaling via dose of labeling antibody or effector to target cell ratio mechanistically impact signaling. For example, such observations could clarify whether the bulk signaling output increases with increasing dose of antibody because a greater proportion of cells are signaling or because the magnitude of signaling

output has increased, or both.

2. Page 16, top paragraph, line 4: It would be helpful to clarify why IVIG facilitates CAR T cell engraftment and why this was necessary for all adaptor injections. Please provide an appropriate citation (the citation listed does not mention this technique).

3. Page 5, Results paragraph 1, line 4: "transcription" should be "transcription".

Response to Reviewers

We thank the reviewers for their positive comments on our revised manuscript and suggestions on how to further improve it. We have conducted additional *in vitro* and *in vivo* experiments and have addressed each reviewer remark in the manuscript as described below in blue.

Reviewer #1 (Remarks to the Author):

Major points:

1. The authors failed to demonstrate the SNAP-CARs being superior to the traditional scFv CAR. Most of the experiments were mainly carried out *in vitro*. The only *in vivo* efficacy experiment was shown in Figure 7. In this experiment, Nalm6 leukemia cells expressing Her2 injected i.v. in NSG mice was used as a model, and an anti-Her2 antibody was used as the adaptor antibody. The experiment should include an anti-CD19 antibody and compare with a CD19 CAR. The advantage of the universal CAR is the flexibility of multiple antigen targeting, but the authors did not demonstrate this *in vivo*.

We appreciate the reviewer's comments, but would also like to point out that the purpose of the research presented in this manuscript is not aiming to be "superior to the traditional scFv CAR" in terms of efficacy, but in flexible targeting capabilities. We are reporting a fundamentally new adaptor CAR that is based on the SNAPtag self-labeling enzyme. The adaptor approach and the unique opportunities that it provides, compared to traditional CARs, are presented in the manuscript.

However, to address this comment and to further demonstrate the flexible targeting capabilities of the SNAP-CAR system *in vivo*, we have now carried out the suggested experiment targeting a second antigen (CD20 instead of CD19 due to antibody accessibility) *in vivo* in the NSG mouse model. In these experiments the SNAP-CAR significantly reduced tumor burden and enhanced survival of mice compared to controls and to a similar level as the traditional anti-CD20 CAR (**Figure S15**). Note, while mice ultimately did succumb to tumor outgrowth, analysis of the tumor cells in the blood of mice showed lack of the targeted CD20 antigen indicating the relapse was due to antigen loss (**Figure S16**).

2. If the authors claim an effect in solid tumors, the cancer model used in Figure 7 should be a solid tumor model instead of a leukemia cell line bearing a solid cancer antigen.

While we target an antigen that is most commonly a solid tumor antigen, and we have interest in ultimately applying the SNAP-CAR system to treat solid tumors, we don't claim here any effect in solid tumors. Of note HER2 is also an antigen on hematological cancers (Oncotarget. 2016 Mar 15; 7(11): 13013–13030). We have chosen the NALM-6

model as it is the gold-standard for CAR T cell development. By engineering NALM-6 to express different target antigens (e.g., HER2, CD20) we are able to benchmark SNAP CAR covalently modified with different antigens/antibodies in a tumor-agnostic manner.

3. There is no characterization of the SNAP-CAR nor SynNotch T cells in vivo. For example, T cell proliferation, infiltration to the tumor sites and T cell persistence. Is the SynNotch/SNAP-CAR system better than the scFv-CARs, as the binding/signaling is different, or are they the same in these aspects?

We thank the reviewers for suggesting additional characterization experiments. We do now include data showing that the SNAP-CAR T cells persist at late time points (for HER2 at least until day 35, the last time point that we had tested for this experiment (**Figure S13**) and for the CD20 experiment until day 40 (**Figure S17**). We also would like to point out that innovations from the SNAP receptors are not directly focused on improving on these aspects compared to traditional CARs, but instead on the programmable targeting of synNotch receptors and CARs as shown in the manuscript.

4. For the cell culture protocols, the authors need to demonstrate their cellular products phenotypes, i.e. Tcm vs Teff phenotypes, and compare these with the traditional scFv-CAR T cell products, such as the Her2-CAR used in this manuscript.

The manuscript included T cell subset data for the CAR T cells including the markers CD4, CD8, and CD62L and show that these are similar for both the SNAP-CAR and the scFv-CAR T cells (**Figure S10**). However, we agree with the reviewer and have now added the additional marker CD45RA to analyze different T cell subsets which we observe to be equivalent between SNAP-CAR T cell and scFv-CAR T cell products (**Figure S14**).

5. Please include other work carried out in this field in the discussion section.

We have now included the following additional descriptions of work in the field of universal adaptor receptors in the discussion section:

“The field of universal receptor engineering is rapidly progressing, including innovations that enhance targeting specificity and adaptor versatility, as well as clinical testing in early-stage trials^{24,62}. To enhance specificity, researchers have developed combinatorial antigen targeting approaches including “AND” logic-based targeting of cells with two antigens on their surface^{17,63}. Additionally, emerging approaches include conditional spatial and temporal control of universal CAR T cells by stimuli such as small molecule drugs and UV light^{64,65}. Further expanding the versatility of universal CARs, researchers have demonstrated targeting via adaptors constructed with molecules beyond IgG antibodies including nanobodies, DARPins, and small molecule drugs^{55,66-68}. Outcomes from several ongoing and planned clinical trials of universal CARs in hematological and solid tumor settings will provide critical information for further development and technological refinement^{62,69}. Finally, pairing the universal targeting capability of adaptor CARs with allogeneic cell approaches and gene editing promises to provide ideal off-the-shelf cell therapeutics⁶⁹.”

Minor points:

1. It's not clear when mentioned "Snap-synNotch cells" in the manuscript in multiple places. i.e. in figure 5, did the authors use PBMC transduced cells or Jurkat cells?

In all cases in the manuscript the SNAP-synNotch receptor is being used in Jurkat cells. We apologize for the confusion and have now further clarified this point in the text saying "SNAP-synNotch Jurkat cells".

Reviewer #2 (Remarks to the Author):

The authors provided a thorough and thoughtful response to points raised in the first round of review. The added experiments, clarification, and minor reorganization included in this revision support key conclusions and strengthen the narrative arc. This revised article is a useful and interesting contribution to the field of synthetic receptor engineering.

We thank the reviewer for these encouraging comments on our revised manuscript.

The following optional suggestions are offered for consideration:

1. Figure 2d: It could be useful to publish single cell-based flow cytometry data of reporter expression as histograms for experiments presented in Figure 2d and Figure S3. This information could elucidate how the proposed strategies to tune receptor signaling via dose of labeling antibody or effector to target cell ratio mechanistically impact signaling. For example, such observations could clarify whether the bulk signaling output increases with increasing dose of antibody because a greater proportion of cells are signaling or because the magnitude of signaling output has increased, or both.

We appreciate the reviewer's insight and have now added flow plots corresponding to Figure 2D for the SNAP-synNotch system paired with the CD19 adaptor (**Figure S3**).

2. Page 16, top paragraph, line 4: It would be helpful to clarify why IVIG facilitates CAR T cell engraftment and why this was necessary for all adaptor injections. Please provide an appropriate citation (the citation listed does not mention this technique).

We apologize for the omission and have now added clarification about the need for IVIG to the manuscript text:

"Of note, intravenous immunoglobulin (IVIG) was administered i.p. on day 4 with CAR T cells and with every adaptor injection to enhance SNAP-CAR T cell engraftment⁴¹. In previous testing, lack of IVIG led to reduction of ~50% of SNAP-CAR T cells in the blood of mice after only 24 hours (Fig 6c. & Supplementary Fig. 11). We hypothesize that the

IVIG is protecting SNAP-CAR cells from Fc Receptor interactions with innate immune cells or stromal cells that are occurring due to lack of circulating antibodies in the NSG mouse model.”

The current citation (Seitz et al. 2021) is indeed correct; however, in the cited manuscript the authors used the trade-name “GammaGuard” instead of IVIG, probably leading to the confusion.

3. Page 5, Results paragraph 1, line 4: “transcription” should be “transcription”.

We have corrected this typo. Thanks for pointing this out.

Reviewers' Comments:

Reviewer #1:

Remarks to the Author:

The authors have fully addressed my questions. Thank you.

Reviewer #2:

Remarks to the Author:

The authors provided a thorough and extensive response to points raised in the last round of review. This manuscript is an excellent contribution to the literature, and no further concerns are noted.